# How solute atoms control aqueous corrosion of Al-alloys

Huan Zhao [1,2] ✉, Yue Yin[1], Yuxiang Wu[1], Siyuan Zhang [1], Andrea M. Mingers[1], Dirk Ponge[1], Baptiste Gault [1,3], Michael Rohwerder [1] & Dierk Raabe [1] ✉

Aluminum alloys play an important role in circular metallurgy due to their good recyclability and 95% energy gain when made from scrap. Their low density and high strength translate linearly to lower greenhouse gas emissions in transportation, and their excellent corrosion resistance enhances product longevity. The durability of Al alloys stems from the dense barrier oxide film strongly bonded to the surface, preventing further degradation. However, despite decades of research, the individual elemental reactions and their influence on the nanoscale characteristics of the oxide film during corrosion in multicomponent Al alloys remain unresolved questions. Here, we build up a direct correlation between the near-atomistic picture of the corrosion oxide film and the solute reactivity in the aqueous corrosion of a high-strength Al-Zn-Mg-Cu alloy. We reveal the formation of nanocrystalline Al oxide and highlight the solute partitioning between the oxide and the matrix and segregation to the internal interface. The sharp decrease in partitioning content of Mg in the peak-aged alloy emphasizes the impact of heat treatment on the oxide stability and corrosion kinetics. Through H isotopic labelling with deuterium, we provide direct evidence that the oxide acts as a trap for this element, pointing at the essential role of the Al oxide might act as a kinetic barrier in preventing H embrittlement. Our findings advance the mechanistic understanding of further improving the stability of Al oxide, guiding the design of corrosion-resistant alloys for potential applications.

Al alloys are widely used in automobiles, ships, and airplanes[1,2], with rapidly growing relevance for weight reduction of (electrical) vehicles, to lower fuel consumption and environmental impact[3]. Besides their excellent strength-to-weight ratio, they are attractive for their high resistance when exposed to corrosive environments, owing to the oxide barriers formed on their surface. These films are very dense and only a few nanometers thick, while their bond to the bulk metal beneath is very strong, inhibiting further corrosion[4,5]. Despite this self-protection mechanism, Al alloys like other metallic alloys, suffer substantial losses as a result of degradation in aqueous environments[6]. Corrosion of Al alloys is both costly and dangerous[7–9]. Three factors deserve particular attention in that context. First, Al alloys are

increasingly exposed to harsh environmental conditions for which they had originally not been designed. Second, alloys are becoming ever stronger and chemically more complex which makes them more vulnerable to corrosion. Third, progress in this field has huge leverage on enhanced sustainability of industrialized societies since it eliminates the need to replace what does not need to be scrapped, a key aspect when enhancing product and infrastructure longevity. Therefore, advancing corrosion protection beyond transient remedial treatment and surface coatings alone towards a more holistic and science-guided corrosion resistant alloy design is becoming increasingly important.

The reaction of Al alloys with aqueous solutions has been of concern since the beginning of the commercial use of Al[10]. The

[1]Max-Planck-Institut für Eisenforschung, Düsseldorf, Germany. [2]State Key Laboratory for Mechanical Behavior of Materials, Xi'an Jiaotong University, Xi'an, China. [3]Department of Materials, Royal School of Mines, Imperial College London, London, UK. ✉e-mail: h.zhao@mpie.de; d.raabe@mpie.de

investigation of the oxide film formed on Al alloys has advanced from macroscopic observations towards more surface-sensitive analysis techniques, e.g. via X-rays photoelectron spectroscopy, Auger electron spectroscopy and Time of Flight Secondary Ion Mass Spectrometry[11–15]. However, the nature of oxidation, metal dissolution, and interface reactions between the oxide film and the metal surface and all the element-specific effects remain elusive. The reason is that the underlying mechanisms are hidden in the nanoscale, and the atomic configurations and partitioning effects at these surface and sub-surface regions with their complex composition and microstructure features are hard to observe. Also, the increasing use of high-strength multi-component Al alloys makes the process of passivation and dealloying much more complex than for pure Al. Yet, when making sustainable Al alloys from scrap, these aspects gain momentum to understand better and define tolerance limits for scrap-related contaminants with respect to their corrosion response[16]. Therefore, a better understanding of the relationship between chemical composition and corrosion could help make a 100 million ton-per-year industry more sustainable.

Here, we address the corrosion mechanisms and the roles of individual elements of a high-strength Al-Zn-Mg-Cu alloy during aqueous corrosion in salt water by combining scanning flow cell (SFC), atom probe tomography (APT), aberration-corrected scanning transmission electron microscopy (STEM) and multicomponent Pourbaix (potential-pH) diagram calculations. For samples of solution heat treated and peak-aged states, we observe the formation of the nanocrystalline (Al,Mg)-rich oxide film and its transition to a detrimental (Zn,Cu)-rich layer from the surface to the interior. We find that the Mg content of the oxide layer decreases from 33 at.% in the solution heat treated state to 15 at.% in the peak-aged sample. Through the use of deuterium (D) we reveal the presence of D (i.e. H) in the electrochemically grown oxide from water splitting, giving hints that the oxide might act as a kinetic barrier in protecting the material from H embrittlement[17,18].

## Results

We studied a 7xxx alloy close to commercial 7050 with a composition of Al–2.69Zn–2.87Mg–0.95Cu–0.05Zr (at.%). The polarization curves for the alloy in the solution heat treated, under-aged, peak-aged, and over-aged conditions, respectively, are shown in Supplementary Fig. 1. The measured open circuit potential of the materials is around $-0.75 \pm 0.05\,V_{Ag/AgCl}$ for all aging states. An Inductively Coupled Plasma-Mass Spectrometer (ICP-MS) combined with a scanning-flow cell was used for time-resolved analysis of the concentration of dissolved atomic ions, as shown in Fig. 1, with the material exposed to 0.01 M KCl in $H_2O$ (see Methods section)[19,20]. ICP-MS offers the unique ability of tracking trace elements at low amounts of total dissolved ions, and we focus here on the dissolution behavior of solute elements Mg, Zn, and Cu in four different aging states, with the dissolution behavior of Al shown in Supplementary Fig. 2. The potential profile applied to the material is shown in Fig. 1a: We imposed a polarization of $-0.6\,V_{Ag/AgCl}$, which is in the anodic potential region for the alloy, for 5 min, followed by a ramp scan to $-1.2\,V$, i.e. into the cathodic region. This potential profile is designed to investigate the dissolution behavior in the anodic regime and its evolution towards cathodic potentials. Figure 1b displays the current densities measured for the material's four different aging states. The positive currents in the anodic potential regime reveal the anodic oxidation and dissolution of the metal, and the steep increase is associated with significant metal dissolution and release rates. The metal species are dissolved at the anodic potential of $-0.6\,V$, and directly past that peak, we observe the strongly decreasing dissolution rates in the cathodic region. A similar overall dissolution behavior is observed for the peak-aged, under-aged, and over-aged alloys, except that less dissolution is observed for the over-aged alloy than for the under-aged and peak-aged states.

Figure 1c–f show the correlated dissolution rates of Zn, Mg, and Cu cations into the electrolyte measured by the mass spectrometer[21,22]. For the solution heat treated alloy, Fig. 1c, we observe a substantial and preferential dissolution of Mg and less Zn dissolution in the active regime, while Cu dissolution is three orders of magnitude lower as shown on the right axis. The dissolution stoichiometry in the active regime reveals significant changes compared to the nominal alloy composition, with the dissolution rate of Mg being 3 times higher, that of Zn being 2 times higher, and the Cu dissolution rate remaining

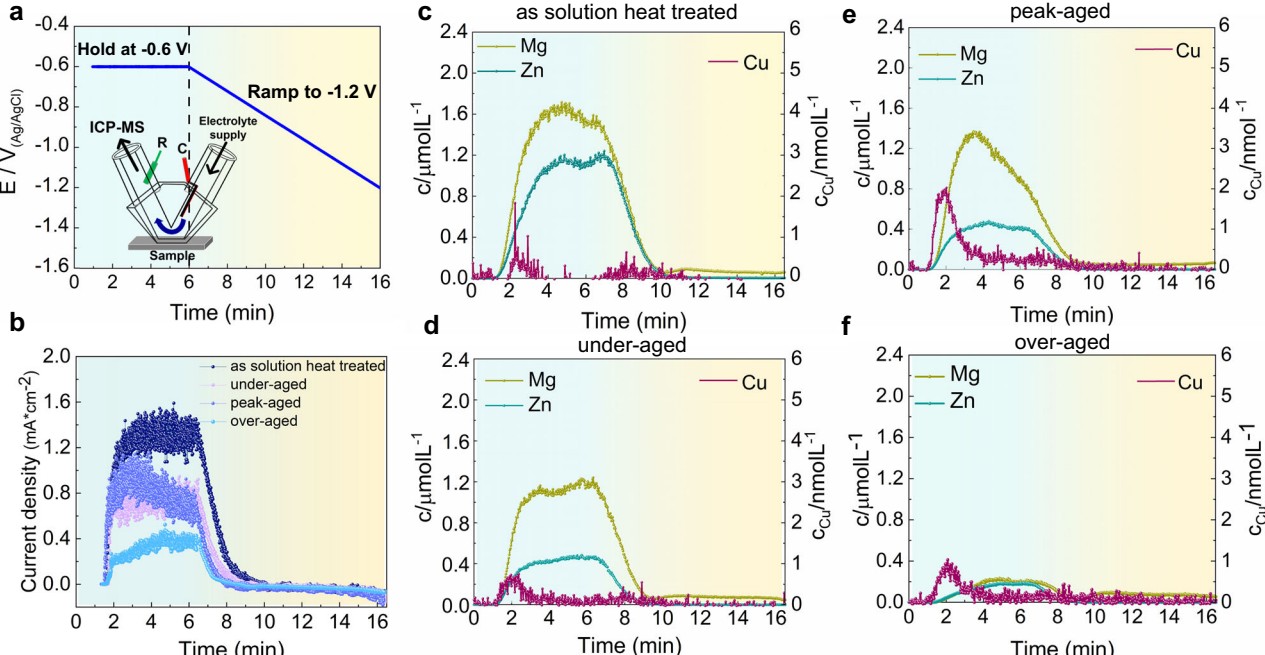

**Fig. 1 | Corrosion properties and element-specific dissolution rates of an AlZnMgCu alloy in 0.01 M KCl. a** The potential applied for the ICP-MS measurement. Inset is a schematic illustration of the ICP-MS. **b** Online ICP-MS dissolution profiles of current densities. **c–f** Online ICP-MS dissolution profiles of dissolved metal ions as a function of time. **c** As solution heat treated (475 °C, 24 h). **d**, Under-aged (120 °C, 2 h). **e** Peak-aged (120 °C, 24 h). **f** Over-aged (120 °C, 24 h + 180 °C, 6 h). Cu is shown on the right axis label.

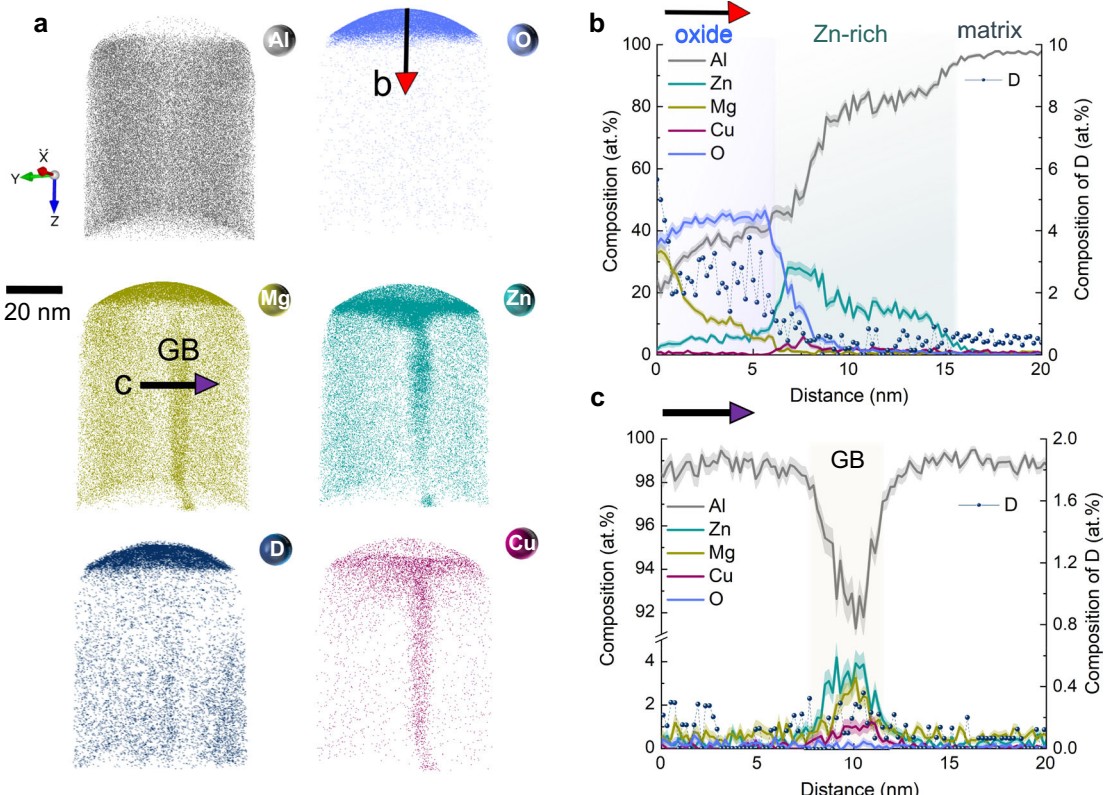

**Fig. 2 | APT analysis on the distribution of individual atoms and composition of the as quenched Al-2.69 Zn-2.87Mg-0.95Cu alloy (at. %) after immersion in 0.1 M KCl in D₂O for 3 h. a** Atom maps of Al, O, Mg, Zn, Cu, and D. **b** One-dimensional composition analysis across the oxide. **c** One-dimensional compositional analysis across the grain boundary. D is shown on the right axis label. The shaded bands of the traces represent the standard deviations in each bin of the composition profile. GB grain boundary.

notably below the bulk composition. The initial dissolution peak of Cu suggests that Cu-rich particles on the surface start to detach and dissolve first[23]. The dissolution behavior of solutes in the peak-aged, under-aged, and over-aged alloys is similar, except that less dissolution of Mg and Zn is observed for the over-aged alloy than for the under-aged and peak-aged states. Al behaves qualitatively as the solutes shown in Supplementary Fig. 2. The lower dissolution rates in the over-aged state can be due to the lower solute contents in the matrix and the stronger electronic bonds of the elements in the precipitates. Also, the coarsening of the precipitates increases their stability in the over-aged state, thus decreasing the driving force for anodic dissolution[24].

The corrosion mechanisms can be better revealed when combining these macroscopic electrochemical dissolution analyses with the characterization of the local chemistry and structure of the oxide formed during the early stage of corrosion. We therefore used APT to resolve the near-atomic scale information of the local chemical distribution near the liquid electrolyte/solid interface in three dimensions at the open circuit potential. The as-solutionized specimens were immersed in a deuterated electrolyte (0.1 M KCl in D₂O) for 3 h. The use of the D isotope instead of H ensures the elimination of detection errors caused by the preparation and intrusion of chamber-acquired H during the analysis[17,25]. APT specimens were prepared to contain both the corroded surface and the underlying uncorroded alloy matrix, using the preparation method given in Supplementary Fig. 3.

Figure 2a presents atom maps of Al, Zn, Mg, Cu, O, and D, with the top pointing towards the corroded surface. A significantly O-rich film is resolved near the surface above the metal matrix. Figure 2b presents the associated compositions of the different regions. O is enriched with up to 40 at.% in the top layer, forming an Al-rich oxide film with a high amount of Mg with up to 33 at.% at the surface. Beneath this (Al,Mg)-rich oxide is a (Zn,Cu)-rich layer with a low O content (5 at.%), i.e. being mainly metallic in nature. Segregation of Zn and Cu is shown at the oxide/metal interface with Zn up to 28 at.% and Cu up to 6 at.%, which is far beyond the solubility limit of Zn (0.85 at.%) and Cu (0.02 at.%)[26] in Al at room temperature. We note that the field change at the oxide/metal interface might induce elemental diffusion at the interface in the APT experiments, a phenomenon that has been reported for light elements such as H, N, C, and O[27,28]. Given the significant amounts observed here of the heavy metals Zn (28 at.%) and Cu (6 at.%), we are inclined to believe that evaporation-induced migration might not have or only very moderately altered the elemental distribution features at the oxide interface. A grain boundary is also observed, with segregation of Zn, Mg, and Cu relative to the matrix, Fig. 2c. The content of O is low, with about 0.3 at.% both in the matrix and at the grain boundary. We characterized the oxide composition in bulk and at the grain boundary, Supplementary Fig. 4, and found no obvious change in the oxide composition. A Mg depletion at the grain boundary next to the surface oxide is shown in Supplementary Fig. 5, which indicates that the grain boundary can enhance the outward diffusion of Mg towards the surface oxide through a short-circuit diffusion mechanism[29].

More interestingly, the atom map and concentration of D show a gradient with a higher content at the surface (5.7 at.%), lower content in the oxide/matrix interface (1.2 at.%), which contrasts with the much lower content of 0.4 at.% in the matrix. A comparison of the mass spectra from the deuterated electrochemical oxide with the samples immersed in KCl in H₂O is shown in Supplementary Fig. 6. The distinct peak at 2 in the specimen immersed in KCl in D₂O together with the comparison with the references of pure Al oxide in Supplementary

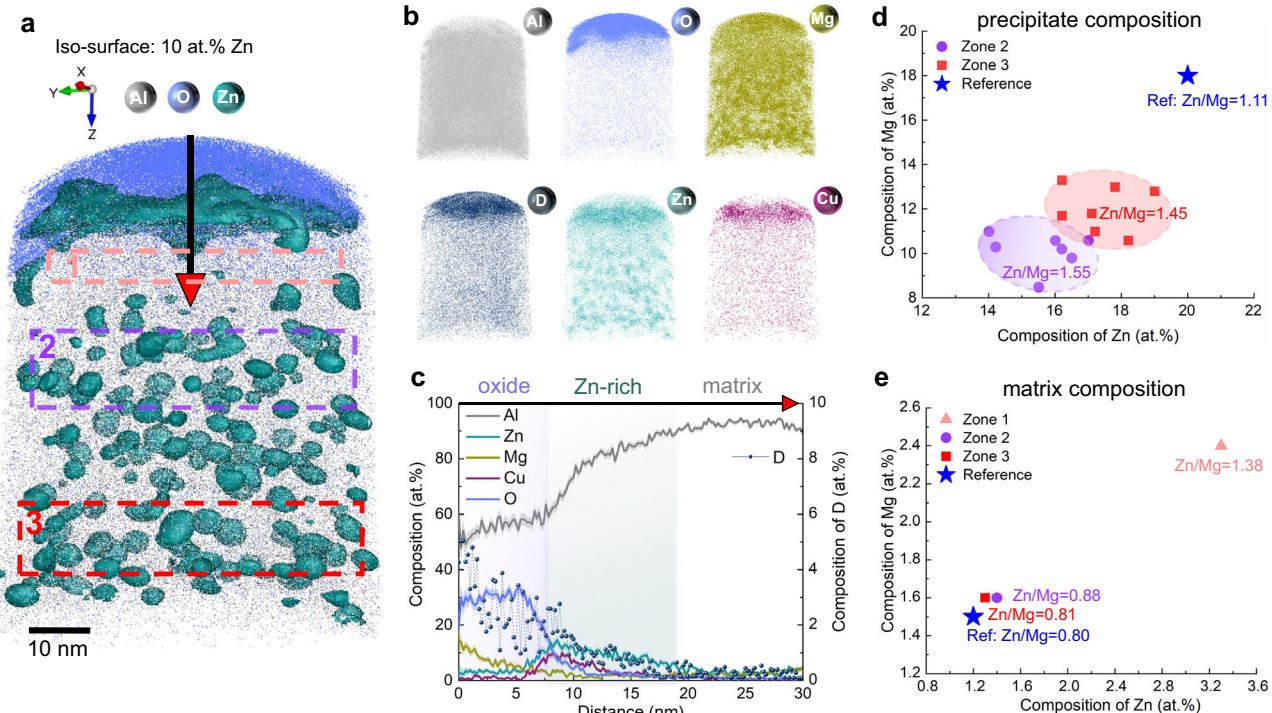

**Fig. 3 | APT analysis of the peak-aged (120 °C, 24 h) Al-2.69 Zn-2.87Mg-0.95Cu alloy (at. %) after immersion in 0.1 M KCl in D$_2$O for 3 h. a** Distribution of O and Al along with iso-surfaces of 10 at.% Zn highlighting the precipitates. **b** Atom maps of Al, O, Mg, D, Zn, and Cu. **c** One-dimensional composition analysis across the oxide. **d**, **e** Precipitate composition and matrix composition with average Zn to Mg ratios analyzed in three neighboring zones, with zones 1–3 marked in pink, purple, and red rectangles in Fig. 3a respectively. These three regions are: zone 1 directly below the oxide where all the former (Mg,Zn)-rich precipitates were dissolved, a transition zone 2 below this region, and zone 3 about 50 nm below the oxide. The reference is referred to the uncorroded sample reported in previous work[37]. The shaded bands of the traces represent the standard deviations in each bin of the composition profile.

Fig. 7 supports the finding of the presence of D in the oxide[17], which indicates that the reaction and evolution of H take place during corrosion[30,31]. The absorbed H can result from the electrolysis of water[11,31] or the hydration of the oxide[32,33]. We refer to the formed O-rich layer simply as oxide throughout for clarity.

A peak-aged sample subjected to the same corrosion condition was analyzed by APT. Figure 3a shows the elemental distribution along with the iso-surfaces that delineate regions containing over 10 at.% Zn in the matrix, highlighting the approximately 5-nm (Mg,Zn)-rich precipitates. Individual atom maps for Al, Zn, Mg, Cu, O and D are also displayed. Similarly to the solution heat treated state, we observe the formation of a (Al,Mg)-rich oxide at the top. Directly beneath is a (Zn,Cu)-rich precipitate-dissolved zone between the oxide and the matrix. Figure 3c shows the compositional profile across the oxide, (Zn,Cu)-rich region, and uncorroded matrix. An increase in Zn and Cu concentrations and depletion of Al is shown directly beneath the oxide.

By comparing with the oxide composition formed in the solution heat treated material in Fig. 2, we see that the Mg content is 55% lower in the peak-aged alloy. Three APT experiments were conducted for each heat treatment and the results show good reproducibility of this difference. The lower Mg content in the oxide in the peak-aged state can be due to the incorporation of Mg into precipitates and its elimination from the solid solution compared to the solution heat treated condition. The grain boundary imaged in Fig. 2 can further enhance the outward diffusion of Mg and increase the Mg content of the surface oxide[29]. The observed (Zn,Cu)-rich layer results from the dissolution of the strengthening precipitates and the consumption of Al and Mg through the formation of the (Al,Mg)-rich oxide. The atom map and concentration of D show that D is enriched up to 5 at.% in the oxide, with the presence of D demonstrated by the distinct D peak in the mass

spectra of the specimen immersed in 0.1 M KCl in D$_2$O in Supplementary Fig. 8[17]. Accurate quantification of O on the surface using APT is a known challenge. This is due to the presence of residual gases in the analysis chamber, potential ingress during sample preparation, and surface contamination during sample transfer and analysis[34–36]. Additionally, the differences in the morphology of the measured surface oxides might also contribute to the variations in surface O content. These factors can account for variations in the measured O concentrations of the surface oxide between Figs. 2 and 3.

To understand the dissolution behavior of precipitates during aqueous corrosion, we analyzed the compositions of precipitates and matrix in three neighboring zones in the peak-aged state, with zones 1–3 marked in pink, purple and red rectangles, respectively, in Fig. 3a. These three regions are: zone 1 directly below the oxide where all the former (Mg,Zn)-rich precipitates were dissolved, a transition zone 2 below this region, and zone 3 about 50 nm below the oxide. The Mg content within the precipitates, shown in Fig. 3d, reduces on average to 10.2 at.% in zone 2 and to 11.0 at.% in zone 3, compared to 18.0 at.% in the uncorroded samples, serving here as a reference value[37]. Zn shows a similar trend, yet with a lower dissolution rate compared to Mg: Zn decreases on average to 16.2 at.% in zone 2 compared to 20 at.% in the reference state. Cu remains almost unchanged in zone 2-3, with a low composition similar to the reference value of 1.6 at.%. The matrix composition in Fig. 3e shows an increase in zone 1 compared to the reference state, which results from the dissolution of precipitates. The matrix probed in zone 2 and 3 shows similar compositions to the reference state. The dissolution of precipitates due to selective oxidation has been shown during high temperature oxidation of superalloys[38,39], and stress corrosion cracking of 7xxx Al alloys[40]. Here we show that the dissolution of precipitates reaches very deep into the material, extending to regions 50 nm below the oxide, with Mg being

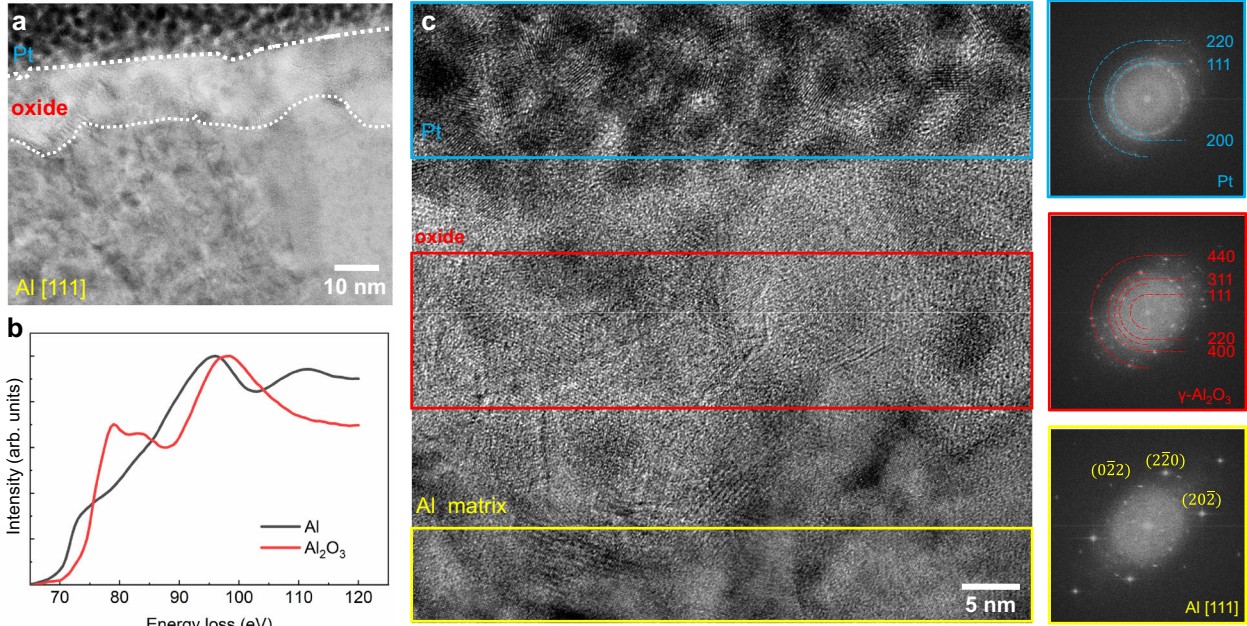

**Fig. 4 | STEM analysis of a peak-aged Al-Zn-Mg-Cu alloy after exposure to 0.1 M KCl for 3 h. a** Low magnification HAADF image showing the cross-section from the corroded surface (covered by Pt) to oxide and Al matrix. **b** STEM-EELS analysis of the oxide and Al matrix. The two EELS spectra are normalized by their maximum counts for a better comparison. **c** FFT patterns obtained from different regions near the oxide.

preferentially dissolved compared to Zn and Cu (see average Zn to Mg ratios in Fig. 3d). The preferential dissolution of Mg of the precipitates is due to the high chemical activity of Mg compared to more noble Zn and Cu upon corrosion[41]. Furthermore, the substantial Mg partitioning (15 at.%) in the oxide provides the driving force for Mg diffusion towards the near-surface region. Mg diffusion from the neighboring matrix and along grain boundaries to the Mg-depleted region near the surface is thus expected, which disturbs the local equilibria leading to the precipitate dissolution and solute diffusion to supply the growth of Mg-rich oxide.

Figure 4a illustrates representative high-resolution images of the oxide in the peak-aged alloy after aqueous corrosion, viewed by bright field (BF) STEM along the <111> zone axis of the Al matrix. The oxide covers the surface, with the thickness estimated to be about 10–20 nm. STEM-Electron Energy Loss Spectroscopy (EELS) was employed to analyze the chemical bonding within the oxide film. Figure 4b shows the near-edge structure of the Al-L$_{2,3}$ edges obtained from metallic Al and the Al-containing oxide. The characteristic of the EELS signal fits well with reported γ-Al$_2$O$_3$ spectra[42]. Figure 4c illustrates different Fast Fourier transform (FFT) patterns that stem from the different regions: these include the Pt protective layer, oxide, and Al matrix. The FFT analysis shows that the oxide has the structure of γ-Al$_2$O$_3$, space group Fd-3m. The variations in FFT patterns observed in Supplementary Fig. 9 further reveal different orientations of each oxide grain within the oxide film, demonstrating that a nanocrystalline structure is found within an amorphous matrix. We also observe substantial density fluctuations in the APT density map of the oxide in Supplementary Fig. 10, a feature which has been reported before as a clear indication for the formation of nanocrystals and grain boundaries[43]. This observation further supports the suggestion that a nanocrystalline oxide structure within an amorphous matrix has formed during corrosion. The oxide formed during the corrosion of Al alloys has been generally considered to be amorphous but was also suggested to transform to a crystalline state over time[10,44,45]. The nature of the nanocrystalline structure of the oxide observed here may offer an optimized surface coverage as it is more compact and stable as opposed to an amorphous structure, which is less ductile and more prone to contain larger

defects[46,47]. Also, the nanocrystalline oxide layer has fast self-healing properties so that it can re-form instantaneously when damaged, thus maintaining the alloy's corrosion protection[47–51]. The STEM and EDS analysis on the oxide in Supplementary Fig. 11 shows that the Zn-rich area beneath the oxide follows the same structure as the Al matrix rather than forming a new Al-Zn phase.

To better understand the reaction mechanism in aqueous corrosion, we performed thermodynamic calculations of the potential-pH diagrams based on the CALculation of the PHAse Diagram method[52]. Using this approach, the oxide phases, main reaction products, and the species present in the aqueous solution or in the solid phases at equilibrium can be predicted for this nominal alloy composition. The multicomponent Pourbaix diagram in Fig. 5 displays the interactions between the Al-Zn-Mg-Cu FCC matrix, oxide, and aqueous solution at given pH and potential values, with the entire range of potential-pH calculations shown in Supplementary Fig. 12. We highlighted the aqueous corrosion condition of the APT and TEM experiments as a black dot in region 2 in Fig. 5a. The γ-Al$_2$O$_3$ is the dominant oxide phase under equilibrium conditions, which is a spinel structure with cation vacancies[10]. Figure 5b shows that the electrochemical corrosion process results in a reduction in the mole fraction of the alloy matrix. Potential-pH diagrams in Fig. 5c, d reveal that Zn and Cu are immune to dissolution in the matrix, while Al preferentially forms γ-Al$_2$O$_3$ and Mg preferentially remains in the aqueous solution as shown in the ion concentrations in Supplementary Fig. 13. The predicted trends agree well with the experimental observations showing Zn enrichment up to 28 at.% and Al decrease in the remaining matrix (Fig. 2b). One has to note though that aqueous corrosion is kinetically limited, and the assumption of local equilibrium concentration at the oxide/metal interface may not hold.

## Discussion

We show that Mg contributes significantly to the formation of Al oxide near the surface during corrosion[53]. Such solute trapping of Mg in Al oxide has been reported previously[40,54]. Both Mg and Al have a very high affinity to oxygen, with Mg having an 8% higher one, explaining its preferential oxidation[54–57]. During the extended oxidation of binary

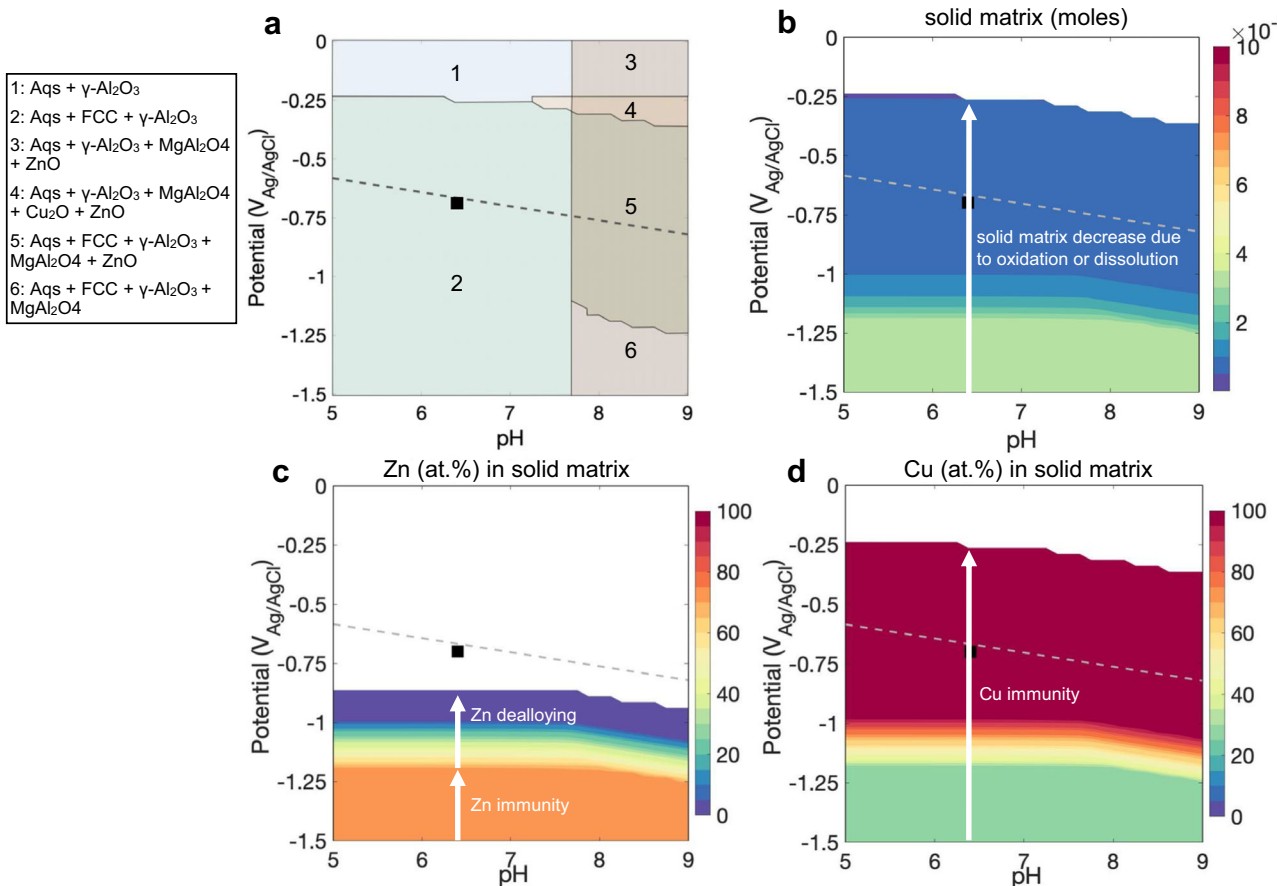

**Fig. 5 | Potential-pH diagrams for the Al-2.69 Zn-2.87Mg-0.95Cu alloy (at. %) in 0.1 M KCl at 25 °C and 1 atm. a** Potential-pH diagram showing equilibrium phase regimes of oxides, aqueous solution (Aqs.), and alloy matrix (FCC). **b–d** Potential-pH diagrams showing the equilibrium dissolution characteristics of matrix phase: **b** Decrease of Al matrix content (moles) as a function of potential and pH. **c**, **d** Equilibrium compositions of Zn and Cu of dealloyed solid matrix. Reversible potentials for H evolution are highlighted as gray dashed lines.

Al-Mg alloys, it was shown that Mg has a strong tendency to segregate and diffuse outward from the matrix developing an Mg-rich oxide near the surface[29,53,58], which is consistent with the Mg enrichment at the surface oxide we observed. However, the oxide that forms with an enhanced Mg content (up to 33 at.%) is not as protective as a pure Al oxide would be. This is due to the fact that Mg-containing oxides are less effective and stable, and may lead to a continuous reaction between salt water and the underlying matrix during corrosion[59]. Pure Al oxide, on the other hand, offers better protection and limits the continued reaction between water and metallic Al below. The Mg-containing Al oxide can thus increase the alloys' general susceptibility to corrosion since it is the most active metal in such engineering materials and corrodes readily in a real environment. In addition, Somjit et al.[60] reported that in a Mg-doped Al oxide, both H solubility and the concentration of H interstitials increase by a factor of $10^4$ and $10^7$ respectively compared to a pure Al oxide. This makes the Mg-doped Al oxides not only much more susceptible to further corrosion but also much less protective against H embrittlement[17].

Moreover, we find that the aging temper affects the composition of the oxide during corrosion, i.e. the Mg content of the surface oxide decreases by 55% of the peak-aged alloy compared to the solution heat treated state. This observation provides access to a strategy for improving the alloy's corrosion resistance by reducing or even eliminating Mg from the surface oxide film[59]. This can be practically achieved by manipulation of the alloy composition and heat treatment with respect to a reduced Mg content of its solid solution content. We indeed observe a lower Mg dissolution rate in the over-aged alloy in

comparison with the peak-aged state in the ICP-MS results. This finding indicates that less Mg content is incorporated within the Al oxide during anodic oxidation and supports our conclusion of better corrosion resistance in the over-aged temper, as shown in STEM results in Supplementary Fig. 14 and reported in the literature[61].

Through isotope-labeling with D, we also show the generation and absorption of D during corrosion[11]. This effect can become essential not only for the corrosion process itself but also for the associated change in the mechanical response of the corroded material, namely, through H-related embrittlement[17,59,62]. The high concentration of D within the Al oxide seems to be due to its efficient trapping within the oxide, preventing further diffusion and adsorption of H into the Al matrix, hence improving the material's resistance to H embrittlement. The D concentrations show a gradient with a substantial enrichment in the more Mg-rich oxide regions. This potentially suggests a correlated high H solubility and diffusivity in this region through the role of Mg on increasing the free H interstitials in Al oxide as elaborated earlier in the theoretical study by Somjit et al.[60]. This points to use a Mg-lean Al oxide rather than the current observed Mg-enriched Al oxide to provide a better performance of Al oxide as H permeation barrier of Al alloys (and other metals).

The more stable solutes Zn and Cu get enriched in the matrix beneath the oxide, which can slow down the progress of corrosion by providing a barrier that inhibits the access of corrosive agents to the metal beneath. However, the enriched layer in the dealloyed region can also create a galvanic cell with the less noble base metal, leading to accelerated corrosion at the interface between the (Zn,Cu)-rich area

and the adjacent alloy matrix[63]. The intrinsic electrochemical instability between the enriched layer and the matrix can also drive the kinetics of localized corrosion, which can significantly deteriorate the alloy's overall corrosion resistance. Further investigations about the local effects of nanoscale composition changes are important for obtaining a better understanding of solute partitioning, trapping, and doping on the stability, structural integrity, and conductivity of the oxide film and the influence of these features on corrosion.

Understanding the underlying mechanisms during aqueous corrosion is one of the grand key missions when aiming to improve the sustainability of the material world, a task particularly challenging in the case of multicomponent engineering alloys. Our study derives specific measures to control and improve the aqueous corrosion resistance of high-strength lightweight Al alloys. One specific strategy aims at enhancing corrosion resistance by reducing the Mg content within the surface Al oxide through manipulation of alloy composition and heat treatment. In light of the observed high H enrichment inside the Al oxide, a viable alloy design and tempering strategy is to promote the formation of pure Al oxide as a permeation barrier that might serve for improving the H embrittlement resistance of the material. This is an important aspect when developing Al-based alloys as H storage media and materials used in transportation and infrastructure for a coming H economy.

## Methods

### Materials
The studied 7xxx Al alloy is with the composition of Al–6.22Zn–2.46Mg–2.13Cu–0.16Zr–0.02Fe in wt.% (Al–2.69Zn–2.87Mg–0.95Cu–0.05Zr–0.01Fe in at.%). The as-cast alloy was homogenized at 475 °C for 24 h and then hot rolled from 40 mm to 3 mm thickness at 450 °C. The hot-rolled alloy was solution heat treated at 475 °C for 24 h with water quenching. Heat treatment was conducted for four conditions[37], solution heat treated (475 °C for 24 h), under-aged (120 °C for 2 h), peak-aged (120 °C for 24 h), and over-aged (120 °C for 24 h + 180 °C for 6 h).

### Electrochemical scanning flow cell
The electrochemical experiments were performed on a micro-electrochemical scanning flow cell (SFC)[21] made of Teflon with a Gamry Reference 600 potentiostat. The elemental dissolution behavior of the above-described four samples was investigated in 0.01 M KCl prepared from Potassium chloride (suprapur, Merck). The measured sample area was approximately 0.01 cm². A Pt-wire counter electrode and a reference electrode (Ag/AgCl/3 M KCl) were positioned in the inlet and outlet channels of the SFC, respectively. An Inductively Coupled Plasma-Mass Spectrometer (ICP-MS) (NexION 300X, Perkin Elmer) was employed for time-resolved measurement of the concentration of dissolved atomic ions ($^{24}$Mg, $^{64}$Zn, $^{63}$Cu). During the measurements, the electrolyte of KCl was pumped with a flow of ca. 200 μL/min into the V-shaped channels of the SFC and further injected into the ICP-MS subsequently. $^{45}$Sc was used for Mg, and $^{74}$Ge for Zn and Cu as standards for detection. An internal standard (standards in HNO₃ and KCl solution) was injected into the electrolyte through a Y-connector behind the SFC to ensure stable performance that corrects for changes in analyte sensitivity caused by variations in the concentration. The potentiostat, electrolyte, gas flows and SFC components were automatically controlled using homemade LabVIEW software[21]. ICP-MS calibration was performed for the elements of interest using the freshly prepared standard solution (i.e. standards in KCl) for background analysis on each measurement day. The detected intensities from the mass spectrometer were calibrated by linear regression, and then the equation of the calibration curve was used to calculate concentrations of dissolved ions in the electrolyte. The dissolution rate of Al is converted by subtracting the solute contribution from the current density applying Faraday's law. $I_{Diss,Me} = \frac{C_{Me} * V_f * Z * F}{A}$,[21]

with $C_{Me}$ being the metal concentration in μmol L$^{-1}$, $V_f$ the flow rate in Ls$^{-1}$, Z the charge number, F the Faraday constant in C mol$^{-1}$ and A the cell area in cm². For Mg and Zn, the charge number is 2$^+$, and Al is 3$^+$. Cu is assumed to be 2$^+$, while Cu is a thousand times lower in the dissolution rate.

### APT/TEM sample preparation
Bulk samples with sizes of 10 mm × 12 mm × 2 mm were exposed to 0.1 M KCl in D₂O (Sigma-Aldrich) for 3 h. The samples were taken out from the electrolyte and placed into the Plasma FIB under a high vacuum of 10$^{-6}$ Torr within less than 5 min. A protective 1 μm Pt layer has been deposited on top of the sample using an electron beam deposition method operated at a low voltage of 10 kV and low current of 1.6 nA, which protected the corroded surface during the entire milling process. Lift-out and sharpening methods using a Si microtip array/Mo grid were used for APT/TEM sample preparation applying the ion beam to the Pt-protected samples. Specimens were prepared and sharpened at 30 kV using a Xe$^+$ beam, followed by a final cleaning step at 5 kV to minimize damage. The APT/TEM sample preparation took approx. 3 h.

### APT analysis
Atom probe experiments were immediately conducted on the prepared specimens using a local electrode atom probe (LEAP 5000XR) with the reflection system. All APT measurements were performed at a temperature of 30 K under ultra-high vacuum of 10$^{-11}$ Torr, using the voltage pulsing mode with a 20% pulse fraction and a 250 kHz pulse rate. APT analysis was performed using the commercial software package AP Suite 6.1. All APT reconstructions were calibrated on the basis of the crystallographic poles observed on the detector hit maps. Three APT experiments were conducted for each heat treatment. The precipitate composition shown in Fig. 3 is obtained from one-dimensional composition analysis. Matrix composition is quantified by measuring the average values over selected areas devoid of precipitates.

### STEM analysis
STEM characterization was conducted on Titan Themis (Thermo Fisher Scientific) microscopes operated at 300 kV. The aberration correction of the probe-forming lenses enabled a STEM probe convergence angle of 24 mrad and a spatial resolution of <0.1 nm. High angle annular dark field and bright field-STEM images were captured employing detectors with collection angles of 73–200 mrad and 0–7 mrad, respectively. Energy dispersive spectroscopy imaging was acquired using the SuperX detector. Electron energy loss spectroscopy (EELS) was performed through a Quantum spectrometer (Gatan). Dual EELS acquisition of the low-loss and high-loss ranges allows for the calibration of zero-loss energy at individual pixels. Multivariate statistical analysis was employed to minimize noise and identify the significant features from the spectrum imaging datasets[64].

### Multicomponent potential-pH calculations
The potential-pH diagrams of the Al–2.69Zn–2.87Mg–0.95Cu (at.%) alloy were calculated based on the CALculation of the PHAse Diagram method using Thermo-Calc software. An interaction system was set between 10$^{-2}$ mol of Al-Zn-Mg-Cu single phase, 0.1 M NaCl in 1 kg of water at 25 °C, and 1 atm ambient pressure. Thermodynamics of the Al matrix, stoichiometric compounds, and aqueous solution are specified by the TCAL5, SSUB5, and TCAQ3 databases in the Thermo-Calc software, respectively. We addressed the oxygen's role in oxidation and corrosion via O-containing ions, excluding dissolved gases, i.e. O₂ and H₂. This is to prevent the dominance of H and O evolution from water electrolysis outside of the stable region of water (indicated by the dashed lines in Fig. 5). This approach preserves the entire stability regions of the oxide phases within the diagram[52]. For the numerical

convergence of the Gibbs energy minimization, we followed the approach by Wang et al.[52] and generated a batch point equilibrium calculation macro to control the POLY-3 module.

## Data availability

All data to evaluate the conclusions are present in the manuscript, and the Supplementary Material. Raw data are available from the corresponding authors on request.

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

## Acknowledgements

We acknowledge Dr. Hao Shi, Dr. Liang Wei, Dr. Baojie Dou, Dr. Mira Todorova, and Prof. Jörg Neugebauer for fruitful discussions. We are grateful to Eric Woods for providing the APT datasets of pure Al oxide for comparison. We greatly acknowledge Philipp Watermeyer for the technical support in the TEM experiment. S.Z. acknowledges the German Research Foundation (DFG) for funding support through SPP 2370 (Project number 502202153).

## Author contributions

H.Z. developed the research concept and carried out the research; D.R. supervised the research project; Y.Y. performed the polarization and OCP tests; A.M.M. conducted ICP-MS measurements; S.Z. performed STEM and EELS measurements; Y.W. performed the multicomponent potential-pH calculations; H.Z. wrote the initial manuscript; D.P., B.G., and M.R. contributed to the scientific discussion of the results and commented on the manuscript.

## Funding

## Competing interests

The authors declare no competing interests.
