## [Peer Review File · Nature Communications]

How solute atoms control aqueous corrosion of Al-alloysREVIEWER COMMENTS

Reviewer #1 (Remarks to the Author):

Comments to authors

The paper focuses on the creation and properties of the oxide of a 7xxx alloy in a KCl solution. The study covers an central topic in metallurgy, and is well executed, with a multi-method approach that surely took a lot of effort to complete. The results are sound and important. While I agree with most of the interpretations of the findings, I have some comments that should be clarified.

- Line 199: Precipitates near the surface lose Mg and Zn. I find this strange. Do they transform to a different phase? It looks like they are shrinking. How was the composition measured? By the isosurfaces? (should be in the experimental section) Could the change in measured composition be an effect of size? I would expect them to stay the same phase but shrink as they dissolve.

- Line 222: I'm not entirely convinced by the EELS spectrum alone as it could also fit other Al oxides, perhaps rephrase to "fits well with reported gamma oxide spectra" or similar. Together with the FFT and thermodynamic calculations, it does make it most likely that the nanocrystals are gamma, perhaps with Mg concentrated in the amorphous part.

- Fig 4: There also seems to be nanocrystals going into the matrix. Or is this still part of the oxide layer? I was hoping to see the form of the Zn segregation below the oxide here, perhaps it creates a crystalline Al-Zn phase. As you have mentioned, the Zn content here is well above the solubility in Al. Any hints from STEM on this?

- Line 291: "This observation provides access to a strategy for improving the alloy's corrosion resistance while maintaining its strength". I doubt this. You will never get all the Mg out of solution in a way that preserves strength, so I think strength should be left out of it. The reference [14] does not seem very relevant.

- Line 303: I would like to see a reference here.

Regarding hydrogen trapping/diffusion (bear with me):

- Line 285: "Somjit et al.49 reported that in a Mg-doped Al oxide, both H solubility and the concentration of H interstitials increase by a factor of 10^4 and 10^7 , respectively. This makes the Mg-doped Al oxides not only much more susceptible to further corrosion but also much less protective against H embrittlement"

- Line 308: "The high concentration of H within the Al oxide seems to be due to its efficient trapping within the oxide, preventing further diffusion and adsorption of H into the Al matrix,"

I read the first quote as "(Al,Mg) oxide can contain a lot of H, therefore H can get through it", and the second quote as "We found a lot of H in (Al,Mg) oxide, therefore H can NOT get through it" (a contradiction). This has to be straightened out.

The way I see it, metal reacts with water and creates an unstable metal hydroxide. When you take it out of the water, most of the hydrogen leaves, and it turns into a more stable metal oxide. As you have measured, some of it is left behind, definitely trapped in the oxide and possibly a little in the matrix. I find this fascinating and important on its own.

But I don't think deuterium being present in the oxide layer can infer whether or not the oxide acts as

a diffusion barrier for H. The deuterium must be strongly bound since it remains through the (non-cryogenic) preparation and APT analysis in high vacuum. But who can say how many interstitial sites are left to fill, and how quickly H could diffuse through them? Speculation is OK, but I would downplay the "kinetic barrier" argument, e.g. in line 27 and 310.

Grammatical/minor comments:

- Line 49: is -> has been.
- Figure 3d,e: Can you please add iso-ratio guide lines, e.g. $Zn/Mg = 1$ or $Zn/Mg = \text{reference}$? The different range of the axes make it hard to see that precipitates are Mg-depleted.
- Line 216: remove the first "the".
- Line 222: oxide scale, what is meant here? The plots are scaled to fit each other, probably? Please rephrase.
- line 284: the -> a.

Reviewer #2 (Remarks to the Author):

This paper presents and discusses the results of a set of experiments conducted to determine the influence on solute elements on the passive film formed on Al-Zn-Mg-Cu alloys, as influenced by heat treatment. The paper is well written, and subject has both scientific and practical implications. However, the following questions/concerns/comments should be addressed prior to publication.

Abstract:

-"However, despite decades of research, the individual elemental reactions and their influence on the nanoscale characteristics of the oxide film during corrosion in multicomponent Al alloys remain unresolved questions" Not sure what the question(s) is(are) that is being addressed here.

Paper:

-"First, Al alloys are increasingly exposed to harsh environmental conditions for which they had originally not been designed" – What has changed with applications in "automobiles ships and airplanes" that substantiates this claim?

-"Second, alloys are becoming ever stronger and chemically more complex which makes them more vulnerable to corrosion" – How so exactly for Al-Zn-Mg-Cu alloys under study.

-What is a "transient remedial treatment"?

-Need to define the "ToF-SIMS" acronym to be consistent with the predominant style used.

-Fig. s1 – would be helpful to include relevant parameters in the caption since they are not included in the Experimental Section. For example, aerated/deaerated/naturally aerated, scan rate and temperature. Are these curves a one-off measurement or ones typical of a replicate data set?

-Fig. 1 – "For the solution heat treated alloy, Fig. 1c, we observe a substantial and preferential dissolution of Mg and less Zn dissolution in the active regime" How much relative to what would be expected if one assumes a stoichiometric dissolution?

-Fig. 2 – It is really hard to see any Mg depletion adjacent to the grain boundary. Perhaps a separate plot of on where the concentration scale is multiplied.

-“By comparing with the oxide composition formed in the solution heat treated material in Fig. 2, we see that the Mg content is 55% lower in the peak-aged alloy” – Could this be a sampling artifact given the fine scale over which the apt tip was extracted? How many tips were examined? Is this difference reproducible?

-“These three regions are: zone 1 directly below the oxide where all the former (Mg,Zn)-rich precipitates were dissolved” – Dissolution from corrosion, not solution heat treatment, correct?
“Here we show that the dissolution of precipitates reaches very deep into the material, extending to regions 50 nm below the oxide, with Mg being preferentially dissolved compared to Zn and Cu (see quantification in Fig. 3d)” – This implies corrosion is occurring sub-surface. Am I understanding this correctly? If true, how can this be?

-“Such solute trapping of Mg in Al oxide has been reported previously^{34,43}” Is it really solute trapping or is it a binary oxide mixture? What is the solubility of MgO in Al₂O₃?
“However, the oxide that forms with an enhanced Mg content (up to 33 at.%) is not as protective as would be a pure Al oxide since Mg continually reacts with water during corrosion⁴⁸” – Mg is oxidized in the oxide, show how does it “continually react”?

-“We indeed observe a lower Mg dissolution rate in the over-aged alloy in comparison with the peak-aged state in the ICP-MS results. This finding indicates that less Mg content is incorporated within the Al oxide during anodic oxidation and supports our conclusion of a better corrosion resistance in the over-aged temper, as also reported in the literature⁵⁰” – Why is the Mg dissolution rate lower? Presumably that is the cause for less Mg incorporation into the oxide in the first place that renders it less protective.

-“The nature of the nanocrystalline structure of the oxide observed here offers an optimized surface¹⁵ coverage as it is more compact and less prone to damage as opposed to an amorphous structure, which is less ductile and more prone to contain larger defects. Also, the nanocrystalline oxide layer has fast self-healing properties so that it can reform instantaneously when damaged, thus maintaining the alloy's corrosion protection.” - These claims are not substantiated by either the experimental results presented or any cited literature.

-“The high concentration of H within the Al oxide seems to be due to its efficient trapping within the oxide, preventing further diffusion and adsorption of H into the Al matrix, hence improving the material's resistance to H embrittlement” – What of the D in the film is a result of hydration (D₂O) rather than in atomic form? I think this claim is a reach based on the lack of evidence presented.

-“However, the enrichment can also lead to local micro-galvanic corrosive attack between the (Zn,Cu)-rich area and the adjacent alloy matrix, triggering pitting corrosion⁵²” – The enrichment looks to be a layer. If true, then how can this be?

-“One specific strategy aims at enhancing corrosion resistance while maintaining the materials' strength by reducing the Mg content within the surface Al oxide through manipulation of alloy composition” – How can this be achieved in a Al-Zn-Mg-Cu alloy?

Experimental Methods:

-The as cast alloy was homogenized at 475 °C and then hot rolled from 40 mm to 3 mm thickness at 450 °C. The hot-rolled alloy was solution heat treated at 475 °C with water quenching” – Please state for how long in each case.

Reviewer #3 (Remarks to the Author):

This is an impressive study combining multiple techniques to look at the corrosion of Al-alloys. It is well-written and the figures are individually very clear, although it does appear that links between results from all the different methodologies are not always apparent. I would like the authors to comment on a number of points before I would be able to recommend publication:

1) In Fig 1 is there any relevant kinetic information which can be extracted from these ICP-MS dissolution profiles? For example, the peak-aged material appears to show quite different behaviour in terms of the shape of the Mg profile compared to as soln treated and under-aged. Furthermore the Cu signal appears to peak first - what do the authors believe is happening here?

2) The oxide compositions in the APT data feel only briefly discussed and compared in some aspects. For example, why is the O signal higher in Fig 2 compared to Fig 3? (~40 vs 30%) Also the fate of Cu seems not mentioned - from the ICP profiles there is a clear dissolution of this - where is it ending up? I can't see a mention of this in the text, only comparison of the sub-surface regions. The atom maps appear to show segregation at the oxide-metal and for Zn, but this is not discussed - also are there any field-evaporation effects influencing the microstructure at this transition stage?

3) Staying with Fig 2, as the authors highlight this is capturing a grain-boundary containing region, which could be expected to demonstrate quite different oxidation behaviour, as the authors note on P10. I'm not convinced therefore this is directly comparable with Fig 3 where the heat-treatment is different. Do the authors not have a non-GB containing APT dataset for the as-quenched to show?

4) The deuterium charging work is certainly novel and of interest - I would however like to see the authors comment on the reproducibility of quantifying this in any one alloy/treatment?

5) The discussion of the STEM data on P11 mentions (for this peak-aged material) that with the oxide 'a nanoscale structure is found within an amorphous matrix'. Then the discussion over P14/15 mentions that for over-aged material 'the nanocrystalline structure...offers an optimised surface coverage as it is more compact and less prone to damage as opposed to an amorphous structure.' I cannot see data in the manuscript to support this - is there a comparable STEM dataset from the over-aged material?

we highly appreciate the most pertinent suggestions from the reviewers. We carefully revised the paper (NCOMMS-23-34381-T entitled "How solute atoms control aqueous corrosion of Al-alloys" along all the points that had been raised. The reviewers' comments are given in black, our replies are in **red** and changes made to the manuscript are highlighted in **yellow**.

REVIEWER COMMENTS

Reviewer #1 (Remarks to the Author):

The paper focuses on the creation and properties of the oxide of a 7xxx alloy in a KCl solution. The study covers **an central topic in metallurgy**, and is **well executed**, with a multi-method approach that surely took a lot of effort to complete. The results are **sound and important**. While I agree with most of the interpretations of the findings, I have some comments that should be clarified.

Answer: We very cordially thank the reviewer for the kind appreciation and the support regarding the scientific quality and importance of this work.

1. - Line 199: Precipitates near the surface lose Mg and Zn. I find this strange. Do they transform to a different phase? It looks like they are shrinking. How was the composition measured? By the isosurfaces? (should be in the experimental section) Could the change in measured composition be an effect of size? I would expect them to stay the same phase but shrink as they dissolve.

Answer: We thank the reviewer for the comment. The composition of the precipitates was obtained by calculating a one-dimensional composition profile in a cylinder along

its length direction. We have now included this description in the Experimental section. Our analysis reveals that precipitates remain in the same phase state but with a slightly smaller size (4-5 nm in diameter) in the region adjacent to the oxide compared to the uncorroded sample (approximately 5-6 nm in diameter). The change in precipitate composition should not be due to size effects since the reduction in composition is quite significant, e.g. Mg reducing from 18.0 at.% in the uncorroded samples to 10.2 at.% in the region adjacent to the oxide. The composition was derived from the one-dimensional profile, which measures the local composition with the cross-sectional area of a cylinder smaller than the precipitate. Furthermore, the composition statistics were obtained from precipitates with different sizes, all of which exhibit lower Mg and Zn contents, which again underpins that the explanation for this phenomenon cannot be size effects (i.e. due to the Gibbs-Thompson relation). Our observation of precipitate dissolution in the near surface regions is similar to related findings that were reported in previous work, where Mg dissolution of precipitates in sub-surface regions following oxidation was also reported ^{1,2}.

2. - Line 222: I'm not entirely convinced by the EELS spectrum alone as it could also fit other Al oxides, perhaps rephrase to "fits well with reported gamma oxide spectra" or similar. Together with the FFT and thermodynamic calculations, it does make it most likely that the nanocrystals are gamma, perhaps with Mg concentrated in the amorphous part.

Answer: We are grateful for the comment. We comply and the sentence has now been rephrased into *"The characteristic of the EELS signal fits well with reported γ - Al_2O_3 spectra."*

3.- Fig 4: There also seems to be nanocrystals going into the matrix. Or is this still part of the oxide layer? I was hoping to see the form of the Zn segregation below the oxide here, perhaps it creates a crystalline Al-Zn phase. As you have mentioned, the Zn content here is well above the solubility in Al. Any hints from STEM on this?

Answer: We thank the reviewer for the comment. The nanocrystals are still part of the oxide. To address the structure analysis of an Zn-rich region, we provide below a STEM and EDS analysis conducted on the oxide. The EDS analysis was challenging due to the significant background from the Pt surroundings and the roughness of the interphase between the oxide and the metal. However, the analysis still clearly reveals Zn segregation beneath the oxide as marked by the yellow arrow. The enrichment is about 5 at.% (Zn/(Al+Zn)) quantified by EDS, which is well above the solubility in Al. In Fig. R1C, we present the FFT analysis for both the matrix and the Zn-rich region. The analysis shows that the Zn-rich area follows the same structure as the Al matrix rather than forming a new Al-Zn phase.

Fig. R1 STEM analysis of a peak-aged Al-Zn-Mg-Cu alloy after exposure to 0.1 M KCl for 3 hours. a, Low magnification HAADF image showing the cross-section from the corroded surface (covered by Pt) to oxide and Al matrix. **b**, STEM-EDS analysis of the oxide and Zn-rich region of the region shown in **a** as indicated by cyan. **c**, FFT patterns obtained from Al matrix and Zn-rich region.

4. - Line 291: "This observation provides access to a strategy for improving the alloy's corrosion resistance while maintaining its strength". I doubt this. You will never get all the Mg out of solution in a way that preserves strength, so I think strength should be left out of it. The reference [14] does not seem very relevant.

Answer: We thank the reviewer for the pertinent and critical suggestion on our previous statement. We fully agree and comply. We remove "while maintaining its strength" from the sentence and add a relevant reference, shown below.

"This observation provides access to a strategy for improving the alloy's corrosion resistance by reducing or even eliminating Mg from the surface oxide film³."

5.- Line 303: I would like to see a reference here.

Answer: References have been added to the manuscript. *“The nature of the nanocrystalline structure of the oxide observed here may offer an optimized surface coverage as it is more compact and stable as opposed to an amorphous structure, which is less ductile and more prone to contain larger defects^{4,5}. Also, the nanocrystalline oxide layer has fast self-healing properties so that it can re-form instantaneously when damaged, thus maintaining the alloy’s corrosion protection⁵⁻⁹.”*

6. Regarding hydrogen trapping/diffusion (bear with me):

- Line 285: "Somjit et al.⁴⁹ reported that in a Mg-doped Al oxide, both H solubility and the concentration of H interstitials increase by a factor of 10^4 and 10^7 , respectively.

This makes the Mg-doped Al oxides not only much more susceptible to further corrosion but also much less protective against H embrittlement".

- Line 308: "The high concentration of H within the Al oxide seems to be due to its efficient trapping within the oxide, preventing further diffusion and adsorption of H into the Al matrix,".

I read the first quote as "(Al,Mg) oxide can contain a lot of H, therefore H can get through it", and the second quote as "We found a lot of H in (Al,Mg) oxide, therefore H can NOT get through it" (a contradiction). This has to be straightened out. The way I see it, metal reacts with water and creates an unstable metal hydroxide. When you take it out of the water, most of the hydrogen leaves, and it turns into a more stable metal oxide. As you have measured, some of it is left behind, definitely trapped in the oxide and possibly a little in the matrix. I find this fascinating and important on its own.

But I don't think deuterium being present in the oxide layer can infer whether or not the oxide acts as a diffusion barrier for H. The deuterium must be strongly bound since it remains through the (non-cryogenic) preparation and APT analysis in high vacuum. But who can say how many interstitial sites are left to fill, and how quickly H could diffuse through them? Speculation is OK, but I would downplay the "kinetic barrier" argument, e.g. in line 27 and 310.

Answer: We appreciate these critical comments and agree that this is a crucial observation, as acknowledged by the reviewer. Experimental difficulties have limited our ability to interpret both the thermodynamic and kinetic effects regarding the H behavior in Mg-containing Al oxide. Our D-labelled measurements clearly confirm the existence of H/D diffusion in the oxide. It is critical to note that while the D level is high in the oxide, the compositional gradient of D decreases immediately upon entering the metal layer. We would like to clarify the following two points:

1. Rate-Limiting Diffusion in the oxide and the matrix: The diffusion inside the oxide is the rate-limiting process and the D activity at the metal/oxide interface controls D diffusion in the metal. This is also intuitively plausible from the standpoint of lattice diffusion theory. Our claim that the oxide serves as a kinetic barrier is a statement in comparison to the metal. We think that this is a valuable information item for the community, as even a passivation layer of a few tens of nanometers can act as a kinetic barrier, a point not to be overlooked in hydrogen embrittlement research.

2. Role of Oxide Chemistry: We observed a multi-component oxide layer, particularly enriched with Mg at the surface oxide. The observed high solubility of D in the oxide is intriguing and prompts us to consider its relationship to Mg enrichment. In the oxide, defect complexes and free H interstitials have implications for both diffusivity and

overall H solubility. Here, we elaborate on the theoretical study submitted earlier by Somjit and Yildiz¹⁰, where the doping strategy has been assessed across Mg, Si, Ti, Fe, and Cr, and it was clearly observed that Mg overwhelmingly stands out for its increased solubility and diffusivity due to its effect on the concentration of free H interstitials. Our analysis of the Mg-enriched oxide shows substantial enrichment of D, suggesting its potential for both high solubility and diffusivity compared to differently-doped oxide variants. This doesn't deny the oxide's role as a kinetic barrier but raises awareness that an even better one is possible than the current investigated Mg-doped Al oxide.

We hope that these aspects clarify the possible contradictions that the reviewer had felt to see from the argumentation chain. We appreciate the reviewer's concerns about the contradiction and overstating the points on H trapping. We have thus modified the text regarding the Mg-doped oxide and downplayed the discussion on the oxide in preventing HE in the manuscript, as shown below.

“The high concentration of D within the Al oxide seems to be due to its efficient trapping within the oxide, preventing further diffusion and adsorption of H into the Al matrix, hence improving the material’s resistance to H embrittlement. The D concentrations show a gradient with a substantial enrichment in the more Mg-rich oxide regions. This potentially suggests a correlated high H solubility and diffusivity in this region through the role of Mg on increasing the free H interstitials in Al oxide as elaborated earlier in the theoretical study by Somjit et al.¹⁰. This points to use a Mg-lean Al oxide rather than the current observed Mg-enriched Al oxide to provide a better performance of Al oxide as H permeation barrier of Al alloys (and other metals).”

*“we provide direct evidence that the oxide acts as a trap for this element, pointing at the essential role of the Al oxide **might act** as a kinetic barrier in preventing H embrittlement.”;*

*“In light of the observed high H enrichment inside the Al oxide, a viable alloy design and tempering strategy is to promote the formation of pure Al oxide as a permeation barrier that **might serve for** improving the H embrittlement resistance of the material.”*

Grammatical/minor comments:

- Figure 3d,e: Can you please add iso-ratio guide lines, e.g. $Zn/Mg = 1$ or $Zn/Mg =$ reference? The different range of the axes make it hard to see that precipitates are Mg-depleted.

Answer: We thank the reviewer for the nice suggestion. We add average Zn to Mg ratios in Fig. 3d,e as shown below.

Fig. 3. APT analysis of the peak-aged (120°C, 24h) Al-2.69 Zn-2.87Mg-0.95Cu alloy (at. %) after immersion in 0.1 M KCl in D₂O for 3 hours. a, Distribution of O and Al along with iso-surfaces of 10 at.% Zn highlighting the precipitates. **b**, Atom maps of Al, O, Mg, D, Zn, and Cu. **c**, One-dimensional composition analysis across the oxide. **d-e**, **Precipitate composition and matrix composition with average Zn to Mg ratios** analyzed in three neighboring zones, with zones 1-3 marked in pink, purple, and red rectangles in Fig. 3a respectively. These three regions are: zone 1 directly below the oxide where all the former (Mg,Zn)-rich precipitates were dissolved, a transition zone 2 below this region, and zone 3 about 50 nm below the oxide. The reference is referred to the uncorroded sample reported in previous work¹¹.

- Line 49: is -> has been.

- Line 216: remove the first "the".

- Line 222: oxide scale, what is meant here? The plots are scaled to fit each other, probably? Please rephrase.

- line 284: the -> a.

Answer: We thank the reviewer for acknowledging this finding. We correct the grammar mistakes. For the oxide scale, we only meant the oxide and we changed it

to “Al containing oxide”. Indeed, the two EELS spectra are normalized by their maximum counts for a better comparison.

We sincerely appreciate the valuable comments from the reviewer, which were indeed most helpful in improving the quality of the manuscript.

Reviewer #2 (Remarks to the Author):

This paper presents and discusses the results of a set of experiments conducted to determine the influence on solute elements on the passive film formed on Al-Zn-Mg-Cu alloys, as influenced by heat treatment. The paper is **well written**, and **subject has both scientific and practical implications**. However, the following questions/concerns/comments should be addressed prior to publication.

We highly appreciate the reviewer's very strong and supporting comments in emphasizing the scientific and practical importance of this work.

Abstract:

1. -"However, despite decades of research, the individual elemental reactions and their influence on the nanoscale characteristics of the oxide film during corrosion in multicomponent Al alloys remain unresolved questions" Not sure what the question(s) is(are) that is being addressed here.

Answer: We thank the reviewer for raising this point. In our paper, we addressed the atomic-scale analysis of the corrosion oxide film and individual elemental reactions of a high-strength Al-Zn-Mg-Cu alloy exposed to aqueous corrosion in salt water. We precisely observe how and where the chemical species are, and particularly how the solutes and precipitates dissolve and react during the corrosion process. We capture the precise local chemical compositions and elemental exchange and partitioning processes occurring between metal and oxide - as well as their changes during corrosion at near-atomic dimensions. In examining another aspect of the atomic-scale corrosion mechanisms, we also discern here for the first time the key role of hydrogen

in that context, which generally arises in humid environments through the process of water splitting occurring during metal corrosion. The distribution of hydrogen within the metal and its oxides has never been directly observed during corrosion at atomic scales, yet grasping its role remains paramount for identifying corrosion-induced failure mechanisms.

Paper:

2. -“First, Al alloys are increasingly exposed to harsh environmental conditions for which they had originally not been designed” – What has changed with applications in “automobiles ships and airplanes” that substantiates this claim?

Answer: We thank the reviewer for this question. Al alloys were initially designed for various applications with specific performance characteristics, but over time, their usage expanded into scenarios characterized by more challenging and harsh environmental conditions. This expansion of their application and product portfolio has been driven by factors such as the need for weight reduction in vehicles and planes paired with the resulting increased efficiency of the products, and the generally increasing demand for mechanically stronger and at the same time corrosion-resistant materials in industries such as the automotive, mobile communication, energy and aerospace sectors. Al alloys are used to reduce the weight of vehicles and improve fuel efficiency. As an example, the figure (taken from the webpage ‘statista’) below shows the drastic increase of the use of Al in vehicles (<https://www.statista.com/statistics/496185/pounds-of-aluminum-per-car-in-north-america/>).

Aluminum consumption in light vehicles in North America from 1975 to 2030

(in net pounds per vehicle)

Fig. R2 Data taken from the webpage 'statista' showing the strong increase of the use of Al in vehicles (<https://www.statista.com/statistics/496185/pounds-of-aluminum-per-car-in-north-america/>).

However, this means that Al components are nowadays indeed subjected to a much wider range of environment and extreme use conditions. Many of the currently high-performing Al alloys with tensile strength values that are reaching and even exceeding the values of some dual-phase steels have been developed over the past 10 years. Their increasing use in demanding applications has necessitated the development of specialized alloys and surface treatments to enhance their resistance to harsh environmental conditions. This can be clearly proven in terms of corresponding patents and publications, as shown in terms of the complex composition in Table 1 and Fig. R3 beneath ^{12,13}. Corrosion in high-strength alloys has resulted in many failures in aircraft structures and components and continues to be a problem today¹⁴.

Table 1. Chemical compositions of some 7xxx Al alloys in wt.%.

Alloy	Zn	Mg	Cu	Zr	Cr	Ti	Mn	Fe	Si
7010	5.7–6.7	2.1–2.6	1.5–2.0	0.10–0.16	<0.05	<0.06	<0.10	<0.15	<0.12
7040	5.7–6.7	1.7–2.4	1.5–2.3	0.05–0.12	<0.04	<0.06	<0.04	<0.13	<0.10
7050	5.7–6.7	1.9–2.6	2.0–2.6	0.10–0.15	<0.04	<0.06	<0.10	<0.15	<0.12
7150	5.9–6.9	2.0–2.7	1.9–2.5	0.08–0.15	<0.04	<0.06	<0.10	<0.15	<0.12
7075	5.1–6.1	2.1–2.9	1.2–2.0	<0.05	0.18–0.28	<0.2	<0.3	<0.5	<0.4
7475	5.2–6.2	1.9–2.6	1.2–1.9	<0.05	0.18–0.25	<0.06	<0.06	<0.12	<0.10
7449	7.5–8.7	1.8–2.7	1.4–2.1	- ^a , ^b	<0.05	- ^a	<0.2	<0.15	<0.10

a
Ti+Zr<0.25.

b
Zr content of 7449 is typically about 0.1 wt%.

Fig. R3 taken from: Raabe et al. Making sustainable aluminum by recycling scrap: The science of “dirty” alloys ¹³.

3. -“Second, alloys are becoming ever stronger and chemically more complex which makes them more vulnerable to corrosion” – How so exactly for Al-Zn-Mg-Cu alloys under study.

Answer: We thank the reviewer for this comment. The chemical, microstructural, and micromechanical complexity of high-strength Al alloys has increased substantially over the past years which makes them more vulnerable to corrosion attack. This is particularly true for alloys of the Al-Zn-Mg-Cu class, which find specific applications in highly demanding aerospace and structural industries due to their superior mechanical properties. The chemical and microstructural complexity of these alloys has been quite drastically increased and the high strength values are mediated by high defect populations such as grain boundaries, coarse grain boundary precipitates, and precipitate-free zones, all of which render materials pertaining to this specific chemically complex alloy class more vulnerable to corrosion-related attack and decay¹⁵. Corrosion in peak-aged Al-Zn-Mg-Cu alloys has resulted in many failure incidents in aircraft structures and components and continues to be a serious and highly safety-critical problem today¹⁴.

4. -What is a “transient remedial treatment”?

Answer: We meant here corrosion prevention measures for maintenance, repair, or replacement of a corroded product. This can provide improved corrosion resistance and make the rate of corrosion lower. However, the treatments are not permanent solutions and may require periodic maintenance or replacement, which causes material decay and failure eventually. This is demonstrated by the annual cost of corrosion representing 4% of the gross national product ¹⁶.

5. -Need to define the “ToF-SIMS” acronym to be consistent with the predominant style used.

Answer: ToF-SIMS is now defined in the manuscript: *“Time of Flight Secondary Ion Mass Spectrometry”*.

6. -Fig. s1 – would be helpful to include relevant parameters in the caption since they are not included in the Experimental Section. For example, aerated/deaerated/naturally aerated, scan rate and temperature. Are these curves a one-off measurement or ones typical of a replicate data set?

Answer: We thank the reviewer for mentioning this. The description has now been added in Fig. S1 as shown below. The polarization curves are representative of three measurements each.

“Fig. S1. Representative polarization curves of Al-Zn-Mg-Cu alloys in deaerated 0.01 M KCl solution with a scan rate of 1 mV/s. As solution heat treated (475 °C, 24 h). Under-aged (120 °C, 2 h). Peak-aged (120 °C, 24 h). Over-aged (120 °C, 24 h+180 °C, 6 h).”

7.-Fig. 1 – “For the solution heat treated alloy, Fig. 1c, we observe a substantial and preferential dissolution of Mg and less Zn dissolution in the active regime” How much relative to what would be expected if one assumes a stoichiometric dissolution?

Answer: We thank the reviewer for pointing this out. The nominal composition of the alloy is 93.44Al–2.69Zn–2.87Mg–0.95Cu–0.05Zr (at.%). The dissolution analysis of the solutionized sample in the active regime yields the dissolution composition: Al: 85.9 %, Mg: 9.0 %, Zn: 5.1 %, Cu: 4×10^{-4} % (molar fractions). The dissolution stoichiometry reveals significant changes in comparison to the bulk composition, with the dissolution rate of Mg being 3 times higher, that of Zn being 2 times higher, and the Cu dissolution rate remaining notably below the bulk composition.

8.-Fig. 2 – It is really hard to see any Mg depletion adjacent to the grain boundary. Perhaps a separate plot of on where the concentration scale is multiplied.

Answer: We thank the reviewer for this comment. The Mg concentration is multiplied

by 5 times and is shown in Supplementary Fig. S5, presented below. We see a clear segregation trend of Mg at the grain boundary far from the surface oxide in Fig.S5c compared with the Mg concentration at the grain boundary next to the surface oxide shown in Fig. S5b.

Fig. S5. Atom probe analysis of the as-quenched Al-Zn-Mg-Cu sample after 3 hours immersion in 0.1M KCl. a, Atom map of Mg. The concentration profiles showing the GB compositions: b, close to the oxide. c, 30 nm below the oxide.

9.-“By comparing with the oxide composition formed in the solution heat treated material in Fig. 2, we see that the Mg content is 55% lower in the peak-aged alloy” – Could this be a sampling artifact given the fine scale over which the apt tip was extracted? How many tips were examined? Is this difference reproducible?

Answer: We thank the reviewer for the question. Three APT experiments were conducted for each heat treatment and the results showed very good reproducibility – we have included another exemplary dataset for each condition in Fig. R4 and Fig. R5.

It shows the consistent observation that the Mg content within the oxide at the surface is around 50% lower in the peak-aged state.

Fig. R4. APT analysis on the atom distribution and composition of the as quenched Al-2.69 Zn-2.87Mg-0.95Cu alloy (at. %) after immersion in 0.1 M KCl solution for 3 hours. a, Atom map containing O, Mg, Zn, and Cu. b, One-dimensional composition analysis across the oxide.

Fig. R5. APT analysis of the peak-aged (120°C, 24h) Al-2.69 Zn-2.87Mg-0.95Cu alloy (at. %) after immersion in 0.1 M KCl solution for 3 hours. a, Distribution of O and Al along with iso-surfaces of 10 at.% Zn highlighting the precipitates. b, One-dimensional composition analysis across the oxide.

10.-“These three regions are: zone 1 directly below the oxide where all the former (Mg,Zn)-rich precipitates were dissolved” – Dissolution from corrosion, not solution heat treatment, correct?

Answer: The reviewer is correct. The corrosion experiment was performed on the peak-aged samples, and the precipitates near the surface were dissolved due to corrosion.

11. “Here we show that the dissolution of precipitates reaches very deep into the material, extending to regions 50 nm below the oxide, with Mg being preferentially dissolved compared to Zn and Cu (see quantification in Fig. 3d)” - This implies corrosion is occurring sub-surface. Am I understanding this correctly? If true, how can this be?

Answer: The results show that corrosion extends to the sub-surface region with preferential dissolution of Mg. The dissolution behavior can be due to the occurrence of oxidation and corrosion during the corrosion test. The preferential dissolution of Mg of the precipitates is due to the high chemical activity of Mg compared to more noble Zn and Cu upon corrosion¹⁷. Furthermore, the Mg partitioning to the oxide with a large content (18 at.%) provides the driving force for Mg diffusion to the near-surface region. Diffusion from the neighboring matrix and along grain boundaries to the Mg-depleted region is expected upon the onset of corrosion, thereby disturbing the local equilibria and causing the dissolution of precipitates to feed the growing Mg-rich oxide.

12. -“Such solute trapping of Mg in Al oxide has been reported previously 34,43” Is it really solute trapping or is it a binary oxide mixture? What is the solubility of MgO in Al₂O₃?

Answer: We thank the reviewer for this question. The STEM-EELS results together with the FFT analysis show the formation of Al₂O₃ oxide, which supports the conclusion of Mg trapping in Al oxide. To further substantiate this, we conducted an X-ray photoelectron spectroscopy (XPS) analysis. The XPS experiment was performed on the sample's surface after immersion under the same previously used corrosive testing conditions. The result is shown in Fig. R6. The O 1s spectrum shows a peak at a binding energy of 530.8 eV, corresponding to the lattice O atoms in Al₂O₃¹⁸, while the Al 2p peak, arising from Al-O-Al, is located at a binding energy of 73.5 eV. The STEM results together with the XPS results show the formation of Al₂O₃ oxide, which supports the conclusion of Mg trapping in Al oxide.

We checked the solubility of MgO in Al₂O₃. The solubility of MgO in Al₂O₃ is very low (in the order of tens of ppm)^{19,20}. These two compounds are more likely to exist as separate phases or distinct materials rather than forming solid solutions with each other. This also suggests that the Mg is in the form of solute trapping Mg.

Fig. R6. XPS spectra of the as quenched Al-2.69 Zn-2.87Mg-0.95Cu alloy (at. %) after immersion in 0.1 M KCl in H₂O for 3 hours. (A) O peak; (B) Al peak.

13. “However, the oxide that forms with an enhanced Mg content (up to 33 at.%) is not as protective as would be a pure Al oxide since Mg continually reacts with water during

corrosion⁴⁸ – Mg is oxidized in the oxide, show how does it “continually react”?

Answer: We appreciate the reviewer for pointing this out. We comply and changed the sentence in the manuscript as shown below:

“However, the oxide that forms with an enhanced Mg content (up to 33 at.%) is not as protective as a pure Al oxide would be. This is due to the fact that Mg-containing oxides are less effective and stable, and may lead to a continuous reaction between salt water and the underlying metal during corrosion³.”

14. -“We indeed observe a lower Mg dissolution rate in the over-aged alloy in comparison with the peak-aged state in the ICP-MS results. This finding indicates that less Mg content is incorporated within the Al oxide during anodic oxidation and supports our conclusion of a better corrosion resistance in the over-aged temper, as also reported in the literature⁵⁰” – Why is the Mg dissolution rate lower? Presumably that is the cause for less Mg incorporation into the oxide in the first place that renders it less protective.

Answer: We thank the reviewer for this question. The lower dissolution rates in the over-aged state can be due to the lower solute contents in the matrix and the stronger electronic bonds of the elements in the precipitates. Also, the coarsening of the precipitates increases their stability in the over-aged state, thus decreasing the driving force for anodic dissolution²¹. The incorporation of Mg into precipitates and its elimination from the solid solution indicates that less Mg gets incorporated into the Al oxide. We propose that pure Al oxide offers better protection. In contrast, the presence of Mg within the Al oxide increases the general susceptibility of the alloys to corrosion since Mg is the chemically most active and least noble element in such complex engineering materials and thus corrodes readily in a real environment.

15.-“The nature of the nanocrystalline structure of the oxide observed here offers an optimized surface coverage as it is more compact and less prone to damage as opposed to an amorphous structure, which is less ductile and more prone to contain larger defects. Also, the nanocrystalline oxide layer has fast self-healing properties so that it can reform instantaneously when damaged, thus maintaining the alloy’s corrosion protection.” - These claims are not substantiated by either the experimental results presented or any cited literature.

Answer: We thank the reviewer for this observation and comment. The relevant reference has now been properly cited.

“The nature of the nanocrystalline structure of the oxide observed here may offer an optimized surface coverage as it is more compact and stable as opposed to an amorphous structure, which is less ductile and more prone to contain larger defects^{4,5}. Also, the nanocrystalline oxide layer has fast self-healing properties so that it can reform instantaneously when damaged, thus maintaining the alloy’s corrosion protection⁵⁻⁹.”

16. “The high concentration of H within the Al oxide seems to be due to its efficient trapping within the oxide, preventing further diffusion and adsorption of H into the Al matrix, hence improving the material’s resistance to H embrittlement” – What of the D in the film is a result of hydration (D₂O) rather than in atomic form? I think this claim is a reach based on the lack of evidence presented.

Answer: We thank the reviewer for this comment. Our STEM analysis utilizing diffraction-based and electron energy loss spectroscopy-based (EELS) methods in Fig. 4 fits well with the γ -Al₂O₃ phase (i.e. not with its hydrated state). We conducted an additional X-ray photoelectron spectroscopy (XPS) analysis on the sample’s surface

after immersion under the same previously used corrosive testing conditions. The result is shown above in Fig. R6. The O 1s spectrum clearly reveals a peak at a binding energy of 530.8 eV, corresponding to the lattice O atoms, while the Al 2p peak, arising from the Al-O-Al bond, is located at a binding energy of 73.5 eV. We also examined the binding signal from 532.5 eV as an indicator of the possible presence of hydrated states¹⁸, but the XPS results do not map or indicate any hydride formation. The composition profile of D in Fig. 2 in the manuscript shows a gradient with a higher content at the surface (5.7 at.%), lower content in the oxide/matrix interface (1.2 at%), and a much lower content in the alloy matrix (0.4 at.%). These collective findings suggest that the oxide acts as the trap for atomic H, i.e., effectively serving as a kinetic barrier for H permeating into the Al matrix. We fully agree and acknowledge that the coexistence with hydroxide phase states may occur under certain environmental conditions and may then also play a role in H embrittlement. We hence comply and downplayed the discussion.

17. “However, the enrichment can also lead to local micro-galvanic corrosive attack between the (Zn,Cu)-rich area and the adjacent alloy matrix, triggering pitting corrosion⁵²” – The enrichment looks to be a layer. If true, then how can this be?

Answer: We thank the reviewer for this comment. The enriched layer in the dealloyed region can create a galvanic cell with the less noble base metal, leading to accelerated corrosion at the interface between this enriched layer and the surrounding metal. The intrinsic electrochemical instability between the enriched layer and the matrix can also drive the kinetics of localized corrosion. This is documented in the work of A. Kosari, where the galvanic interaction is found to be enhanced between the Cu-rich layer and the adjacent alloy matrix, ultimately resulting in the dissolution of the alloy matrix in an Al-Cu alloy during corrosion as revealed by quasi in-situ TEM approach²². We

acknowledge that Zn and Cu might also contribute to the formation of a protective barrier layer on the surface of the alloy during the later stage of corrosion. This barrier layer can slow down the progress of corrosion by providing a barrier that inhibits the access of corrosive agents to the underlying metal. We discussed both aspects in the manuscript for a more thorough understanding.

18. “One specific strategy aims at enhancing corrosion resistance while maintaining the materials’ strength by reducing the Mg content within the surface Al oxide through manipulation of alloy composition” – How can this be achieved in a Al-Zn-Mg-Cu alloy?

Answer: We would like to thank the reviewer for this comment. As also commented by the first reviewer it is hard to improve both the corrosion resistance and maintain its strength at the same time. We thus comply and removed the statement item “...maintain its strength..” from the discussion. In order to enhance the corrosion resistance of Al alloys, reducing the Mg content within the surface Al oxide is crucial, as demonstrated in our study. This can be achieved by practical methods, such as manipulating the alloy composition, e.g. adjusting the Zn to Mg ratios and adding alloying elements, in order to trap Mg into the precipitates and less in solid solution. Additionally, heat treatment can be employed to manipulate the precipitation kinetics, e.g. by applying multi-step aging treatments.²³

19. Experimental Methods:

-The as cast alloy was homogenized at 475 °C and then hot rolled from 40 mm to 3 mm thickness at 450 °C. The hot-rolled alloy was solution heat treated at 475 °C with water quenching” – Please state for how long in each case.

Answer: The experimental details have now been added to the manuscript.

“The as-cast alloy was homogenized at 475 °C for 24h and then hot rolled from 40 mm to 3 mm thickness at 450 °C. The hot-rolled alloy was solution heat treated at 475 °C for 24h with water quenching.”

We sincerely appreciate this valuable comment from the reviewer, which is very helpful in improving the quality of the manuscript.

Reviewer #3 (Remarks to the Author):

This is an **impressive** study combining multiple techniques to look at the corrosion of Al-alloys. It is well-written and the figures are individually very clear, although it does appear that links between results from all the different methodologies are not always apparent. I would like the authors to comment on a number of points before I would be able to recommend publication:

We very cordially thank the reviewer for the kind appreciation and the support regarding the scientific quality of this work.

1) In Fig 1 is there any relevant kinetic information which can be extracted from these ICP-MS dissolution profiles? For example, the peak-aged material appears to show quite different behaviour in terms of the shape of the Mg profile compared to as soln treated and under-aged. Furthermore, the Cu signal appears to peak first - what do the authors believe is happening here?

Answer: We cordially thank the reviewer for this valuable question. The ICP-MS dissolution profiles illustrate the **dynamic** dissolution rates of Zn, Mg, and Cu cations into the electrolyte as a function of time, as measured by the mass spectrometer. Hence, it would not be prudent to draw conclusions regarding the relationship between the shape changes of Mg and its dissolution behavior.

We acknowledge the reviewer for the question regarding the dissolution behavior of Cu. The peak shows that Cu dissolution commenced when the sample was exposed to the electrolyte, with dissolution rates reaching approximately $C_{Cu} = 1$ nmol/L. Notably, following the initial peak, Cu dissolution remains stable during the experiment duration

at a very low rate with 1000 times lower than that of Mg and Zn. This observation suggests that Cu-rich particles on the surface started to detach and dissolve first²⁴. We incorporate a discussion on Cu dissolution on Page 5 in the manuscript.

2) The oxide compositions in the APT data feel only briefly discussed and compared in some aspects. For example, why is the O signal higher in Fig 2 compared to Fig 3? (~40 vs 30%) Also the fate of Cu seems not mentioned - from the ICP profiles there is a clear dissolution of this - where is it ending up? I can't see a mention of this in the text, only comparison of the sub-surface regions. The atom maps appear to show segregation at the oxide-metal and for Zn, but this is not discussed - also are there any field-evaporation effects influencing the microstructure at this transition stage?

Answer: We thank the reviewer for raising this question. Accurate quantification of O on the surface using APT is a known challenge. This is due to the presence of residual gases in the analysis chamber, potential ingress during sample preparation, and surface contamination during sample transport and analysis²⁵⁻²⁷. Additionally, the differences in the morphology of the measured surface oxides might also contribute to the variations in the surface O content. These factors can account for variations in the measured O concentrations of the surface oxide between the two datasets. We incorporated this information on Page 8 in the manuscript.

We discussed the segregation of Zn and Cu on the metal-oxide interface in the manuscript as shown below. *“Beneath this (Al,Mg)-rich oxide is a (Zn,Cu)-rich layer with a low O content (5 at.%), i.e. being mainly metallic in nature. Segregation of Zn and Cu is shown at the oxide/metal interface with Zn up to 28 at.% and Cu up to 6 at.%, which is far beyond the respective solubility limits of Zn (0.85 at.%) and Cu (0.02 at.%)²⁸ in Al at room temperature.”*

“The observed (Zn,Cu)-rich layer results from the dissolution of the strengthening precipitates and the consumption of Al and Mg through the formation of the (Al,Mg)-rich oxide.”

We thank the reviewer for expressing concerns about the field evaporation effects on the measurement at the oxide/metal interface. This migration effect has indeed been reported for light elements such as H, N, C, and O^{29,30}. The measurements were all conducted in voltage mode, an approach that does not lead to such artifacts for the heavy metals of Zn and Cu. We also discussed the potential field evaporation effects on Page 7 in the manuscript as shown below.

“We note that the field change at the oxide/metal interface might induce elemental diffusion at the interface in the APT experiments. This migration effect has been reported for light elements such as H, N, C, and O^{29,30}. Given the significant amounts observed here of the heavy metals Zn and Cu and the fact that the measurements have been conducted in voltage mode, we believe that evaporation-induced migration has not or only very moderately altered the elemental distribution features at the oxide interface.”

3) Staying with Fig 2, as the authors highlight this is capturing a grain-boundary containing region, which could be expected to demonstrate quite different oxidation behaviour, as the authors note on P10. I'm not convinced therefore this is directly comparable with Fig 3 where the heat-treatment is different. Do the authors not have a non-GB containing APT dataset for the as-quenched to show?

Answer: We thank the reviewer for asking this question. We present below an APT dataset of the as-quenched state containing no grain boundary. We observe that the Mg content within the oxide in the as-quenched state is approximately 50% higher

than in the peak-aged state shown in Fig. 3 in the main manuscript. This observation is consistent with the discussion on the influence of heat treatment in the partitioning content of Mg in the oxide.

Fig. R7. APT analysis on the atom distribution and composition of the as quenched Al-2.69 Zn-2.87Mg-0.95Cu alloy (at. %) after immersion in 0.1 M KCl in D₂O for 3 hours. a, Atom map containing O, Mg, Zn, and Cu. b, One-dimensional composition analysis across the oxide.

4) The deuterium charging work is certainly novel and of interest - I would however like to see the authors comment on the reproducibility of quantifying this in any one alloy/treatment?

Answer: We thank the reviewer for the comment on the reproducibility. In Fig. S7 in the manuscript, we discussed the reliability of D measurements within the oxide in APT. We compared the ratios of 1 Da (H⁺) to 2 Da (H₂⁺ and/or D) against local electrical field strength (for this we use the ratio of Al²⁺/Al⁺ as a proxy, based on Kingham's theory for post-ionization³¹, within the oxide of the sample corroded in 0.1 M KCl in D₂O. **5 datasets on the quantification of D in the oxides from the current study were shown.** Reference values include measurements on the uncorroded reference alloy, oxide within the sample corroded in 0.1 M KCl in H₂O, and datasets

taken on bulk Al oxide samples. The comparison shows a significant decrease with an order of magnitude of the 1 Da/2 Da ratios in the 5 datasets compared to reference values. These analyses indicate that the peak at 2 Da in the samples corroded in 0.1 M KCl in D₂O is extremely unlikely to be associated with H₂⁺ but with D, thereby supporting our interpretation.

Fig. S7. Reliability of D measurements within the oxide in APT. The ratios of 1 Da (H⁺) to 2 Da (H₂⁺ and/or D) within the oxide of the sample corroded in 0.1 M KCl in D₂O plotted against field strength, showing an increase with an order of magnitude of the D composition compared to reference values. Reference values include measurements on uncorroded alloy, oxide within the sample corroded in 0.1 M KCl in H₂O, and in a pure Alumina.

5) The discussion of the STEM data on P11 mentions (for this peak-aged material) that with the oxide 'a nanocrystalline structure is found within an amorphous matrix'. Then the discussion over P14/15 mentions that for over-aged material 'the nanocrystalline structure...offers an optimised surface coverage as it is more compact and less prone to damage as opposed to an amorphous structure.' I cannot see data

in the manuscript to support this - is there a comparable STEM dataset from the over-aged material?

Answer: We thank the reviewer for this question. Indeed, both sentences are intended to show the nanocrystalline oxide within an amorphous matrix in the peak-aged alloy observed in Fig. 4. The first one on P11 was the description of Fig. 4 and the second sentence in the discussion on P14 was intended to discuss the relationship of the observed nanocrystalline oxide with corrosion behavior. This is supported by the references. We have thus moved the discussion on P14 to Page 11 for clarity.

“The nature of the nanocrystalline structure of the oxide observed here may offer an optimized surface coverage as it is more compact and stable as opposed to an amorphous structure, which is less ductile and more prone to contain larger defects^{4,5}. Also, the nanocrystalline oxide layer has fast self-healing properties so that it can reform instantaneously when damaged, thus maintaining the alloy’s corrosion protection⁵⁻⁹.”

We have also conducted STEM analysis on the over-aged alloy after immersion under the same corrosive testing conditions previously used for the peak-aged alloy. The results presented in the figure below show the cross-section from the corroded surface (covered by Pt) to the oxide and the Al matrix. We can see that the over-aged material exhibits a more uniform and compact oxide formation, with less corrosion occurrence compared to the peak-aged sample. This finding supports the discussion that the over-aged material shows better corrosion resistance.

Fig. R8. STEM analysis of Al-Zn-Mg-Cu alloy after exposure to 0.1 M KCl solution for 3 hours. a, Peak-aged b, Over-aged.

We appreciate the suggestions from the reviewer and sincerely hope that the reviewer is satisfied with the response and the associated version.

References

- 1 Schwarzenböck, E. *et al.* Evolution of surface characteristics of two industrial 7xxx aluminium alloys exposed to humidity at moderate temperature. *Surface and Interface Analysis* **51**, 1288-1297 (2019).
- 2 López Freixes, M. *et al.* Revisiting stress-corrosion cracking and hydrogen embrittlement in 7xxx-Al alloys at the near-atomic-scale. *Nature Communications* **13**, 4290 (2022).
- 3 Scully, J., Young Jr, G. & Smith, S. in *Gaseous hydrogen embrittlement of materials in energy technologies* 707-768 (Elsevier, 2012).
- 4 Xia, Z. X., Zhang, C., Huang, X. F., Liu, W. B. & Yang, Z. G. Improve oxidation resistance at high temperature by nanocrystalline surface layer. *Scientific Reports* **5**, 13027 (2015).
- 5 Marcus, P. Surface science approach of corrosion phenomena. *Electrochimica Acta* **43**, 109-118 (1998).
- 6 Marcus, P. & Maurice, V. in *Materials Science and Technology: A Comprehensive Treatment* 131-169 (2000).
- 7 Kwok, C. T., Cheng, F. T., Man, H. C. & Ding, W. H. Corrosion characteristics of nanostructured layer on 316L stainless steel fabricated by cavitation-annealing. *Materials Letters* **60**, 2419-2422 (2006).

- 8 Maurice, V. & Marcus, P. Current developments of nanoscale insight into corrosion protection by passive oxide films. *Current Opinion in Solid State and Materials Science* **22**, 156-167 (2018).
- 9 Gialanella, S. & Malandrucolo, A. in *Aerospace Alloys* 439-499 (Springer International Publishing, 2020).
- 10 Somjit, V. & Yildiz, B. Doping α -Al₂O₃ to reduce its hydrogen permeability: Thermodynamic assessment of hydrogen defects and solubility from first principles. *Acta Materialia* **169**, 172-183 (2019).
- 11 Zhao, H., Gault, B., Ponge, D., Raabe, D. & De Geuser, F. Parameter free quantitative analysis of atom probe data by correlation functions: Application to the precipitation in Al-Zn-Mg-Cu. *Scripta Materialia* **154**, 106-110 (2018).
- 12 Starink, M. J. & Wang, S. C. A model for the yield strength of overaged Al-Zn-Mg-Cu alloys. *Acta Materialia* **51**, 5131-5150 (2003).
- 13 Raabe, D. *et al.* Making sustainable aluminum by recycling scrap: The science of “dirty” alloys. *Progress in Materials Science* **128**, 100947 (2022).
- 14 Knight, S. P., Birbilis, N., Muddle, B. C., Trueman, A. R. & Lynch, S. P. Correlations between intergranular stress corrosion cracking, grain-boundary microchemistry, and grain-boundary electrochemistry for Al-Zn-Mg-Cu alloys. *Corrosion Science* **52**, 4073-4080 (2010).
- 15 Rao, A. C. U., Vasu, V., Govindaraju, M. & Srinadh, K. V. S. Stress corrosion cracking behaviour of 7xxx aluminum alloys: A literature review. *Transactions of Nonferrous Metals Society of China* **26**, 1447-1471 (2016).
- 16 Landolt, D. *Corrosion and surface chemistry of metals*. (CRC press, 2007).
- 17 Guillaumin, V. & Mankowski, G. Localized corrosion of 2024 T351 aluminium alloy in chloride media. *Corrosion Science* **41**, 421-438 (1998).
- 18 Kumar, N. & Biswas, K. Cryomilling: An environment friendly approach of preparation large quantity ultra refined pure aluminium nanoparticles. *Journal of Materials Research and Technology* **8**, 63-74 (2019).
- 19 Miller, L., Avishai, A. & Kaplan, W. D. Solubility limit of MgO in Al₂O₃ at 1600° C. *Journal of the American Ceramic Society* **89**, 350-353 (2006).
- 20 Ando, K. & Momoda, M. Solubility of MgO in single-crystal Al₂O₃. *J. Ceram. Soc. Jpn* **95**, 381-386 (1987).
- 21 Goswami, R., Lynch, S., Holroyd, N., Knight, S. P. & Holtz, R. L. Evolution of grain boundary precipitates in Al 7075 upon aging and correlation with stress corrosion cracking behavior. *Metallurgical and Materials Transactions A* **44**, 1268-1278 (2013).
- 22 Kosari, A. *et al.* Dealloying-driven local corrosion by intermetallic constituent particles and dispersoids in aerospace aluminium alloys. *Corrosion Science* **177**, 108947 (2020).
- 23 Azarniya, A., Taheri, A. K. & Taheri, K. K. Recent advances in ageing of 7xxx series aluminum alloys: A physical metallurgy perspective. *Journal of Alloys and Compounds* **781**, 945-983 (2019).
- 24 Gharbi, O. *et al.* On the corrosion of additively manufactured aluminium alloy AA2024 prepared by selective laser melting. *Corrosion Science* **143**, 93-106 (2018).
- 25 Lozano-Perez, S., Saxey, D. W., Yamada, T. & Terachi, T. Atom-probe tomography characterization of the oxidation of stainless steel. *Scripta Materialia* **62**, 855-858 (2010).

- 26 Kinno, T., Tomita, M., Ohkubo, T., Takeno, S. & Hono, K. Laser-assisted atom probe tomography of ^{18}O -enriched oxide thin film for quantitative analysis of oxygen. *Applied Surface Science* **290**, 194-198 (2014).
- 27 Pérez-Huerta, A., Laiginhas, F., Reinhard, D. A., Prosa, T. J. & Martens, R. L. Atom probe tomography (APT) of carbonate minerals. *Micron* **80**, 83-89 (2016).
- 28 Dirks, A. G. & van den Broek, J. J. θ - Al_2Cu formation at room temperature in metastable Al–Cu alloy films. *Acta Metallurgica* **37**, 9-15 (1989).
- 29 Bachhav, M. *et al.* Interpreting the Presence of an Additional Oxide Layer in Analysis of Metal Oxides–Metal Interfaces in Atom Probe Tomography. *The Journal of Physical Chemistry C* **123**, 1313-1319 (2019).
- 30 McCarroll, I., Scherrer, B., Felfer, P., Moody, M. P. & Cairney, J. M. Interpreting atom probe data from oxide–metal interfaces. *Microscopy and Microanalysis* **24**, 342-349 (2018).
- 31 Gault, B., Moody, M. P., Cairney, J. M. & Ringer, S. P. *Atom probe microscopy*. Vol. 160 (Springer Science & Business Media, 2012).

REVIEWERS' COMMENTS

Reviewer #1 (Remarks to the Author):

The reviewed manuscript has addressed the concerns of all reviewers and is greatly approved. I recommend acceptance and look forward to its publication.

Reviewer #2 (Remarks to the Author):

The authors have provided thoughtful Responses to all of comments and questions raised by the Reviewers. While I would have liked to see more of the responses included in the revision, it is suitable for publication.

Reviewer #3 (Remarks to the Author):

The authors have responded to these comments appropriately; just note however concerning artefacts in APT, it is not the case that running in voltage mode will remove these compared to laser mode; often there are still residual issues particularly when moving into regions with differing evaporation fields.

The authors should therefore adjust amended text to avoid making this assertion.

REVIEWERS' COMMENTS

Reviewer #1 (Remarks to the Author):

The reviewed manuscript has addressed the concerns of all reviewers and is greatly approved. I recommend acceptance and look forward to its publication.

Answer: We cordially thank the reviewer for their kind appreciation of the revision work.

Reviewer #2 (Remarks to the Author):

The authors have provided thoughtful Responses to all of comments and questions raised by the Reviewers. While I would have liked to see more of the responses included in the revision, it is suitable for publication.

Answer: We highly appreciate the reviewer's support. We thank the reviewer for the comment on incorporating the responses into the manuscript. We fully comply and include the responses in the revised manuscript where applicable.

Page 6: "For the solution heat treated alloy, Fig. 1c, we observe a substantial and preferential dissolution of Mg and less Zn dissolution in the active regime, while Cu dissolution is three orders of magnitude lower as shown on the right axis. The dissolution stoichiometry in the active regime reveals significant changes compared to the nominal alloy composition, with the dissolution rate of Mg being 3 times higher, that of Zn being 2 times higher, and the Cu dissolution rate remaining notably below the bulk composition."

Page 10: "Three APT experiments were conducted for each heat treatment and

the results show good reproducibility of this difference.”

Page 12: “Here we show that the dissolution of precipitates reaches very deep into the material, extending to regions 50 nm below the oxide, with Mg being preferentially dissolved compared to Zn and Cu (see average Zn to Mg ratios in Fig. 3d). The preferential dissolution of Mg of the precipitates is due to the high chemical activity of Mg compared to more noble Zn and Cu upon corrosion⁴¹. Furthermore, the substantial Mg partitioning (15 at.%) in the oxide provides the driving force for Mg diffusion towards the near-surface region. Mg diffusion from the neighboring matrix and along grain boundaries to the Mg-depleted region near the surface is thus expected, which disturbs the local equilibria leading to the precipitate dissolution and solute diffusion to supply the growth of Mg-rich oxide.”

Page 16: “The more stable solutes Zn and Cu get enriched in the matrix beneath the oxide, which can slow down the progress of corrosion by providing a barrier that inhibits the access of corrosive agents to the metal beneath. However, the enriched layer in the dealloyed region can also create a galvanic cell with the less noble base metal, leading to accelerated corrosion at the interface between the (Zn,Cu)-rich area and the adjacent alloy matrix⁶². The intrinsic electrochemical instability between the enriched layer and the matrix can also drive the kinetics of localized corrosion, which can significantly deteriorate the alloy’s overall corrosion resistance.”

Reviewer #3 (Remarks to the Author):

The authors have responded to these comments appropriately; just note however concerning artefacts in APT, it is not the case that running in voltage mode will remove these compared to laser mode; often there are still residual issues particularly when moving into regions with differing evaporation fields. The authors should therefore adjust amended text to avoid making this assertion.

Answer: We cordially thank the reviewer for their appreciation of our revision work. We appreciate the reviewer's comment about the APT results. We have thus modified the text and downplayed the discussion, as shown below.

“We note that the field change at the oxide/metal interface might induce elemental diffusion at the interface in the APT experiments, a phenomenon that has been reported for light elements such as H, N, C, and O^{27,28}. Given the significant amounts observed here of the heavy metals Zn (28 at.%) and Cu (6 at.%) ~~and the fact that the measurements have been conducted in voltage mode,~~ we are inclined to believe that the evaporation-induced migration might not have or only very moderately altered the elemental distribution features at the oxide interface.”